# Nutrient transport and transformation in macrotidal estuaries of the French Atlantic coast: a modelling approach using C-GEM

Xi Wei[1*], Josette Garnier[1*], Vincent Thieu[1], Paul Passy[2], Romain Le Gendre[3], Gilles Billen[1], Maia Akopian[4], and Goulven Gildas Laruelle[5]

1 UMR Metis 7619, Sorbonne Université, CNRS EPHE, 4 Place Jussieu, Paris, 75005, France
2 UMR 8586 PRODIG, Université de Paris, 8 rue Albert Einstein, 75013 Paris, France
3 Ecosystèmes et Aquaculture Durable, Unité de Recherche Lagons, IFREMER, Nouméa, 98897, New-Caledonia
4 Department of Research and Scientific Support, French Biodiversity Agency (OFB), 5 square Félix Nadar, 94300 Vincennes, France
5 Department of Geosciences, Environment and Society, Université Libre de Bruxelles, Brussels, 1050, Belgium

*Correspondence to:* Xi Wei (xi.wei@upmc.fr; xi.wei_fr@hotmail.com); Josette Garnier (josette.garnier@upmc.fr).

**Abstract:** Estuaries are key reactive ecosystems along the land–ocean aquatic continuum, with significant ecological and economic value. However, they have been facing strong morphological management changes as well as increased nutrient and contaminant inputs, possibly leading to ecological problems such as coastal eutrophication. Therefore, it is necessary to quantify the import and export fluxes of the estuaries, their retention capacity, and estuarine eutrophication potential. The 1-D Carbon–Generic Estuary Model (C-GEM) was used to simulate the transient hydrodynamics, transport, and biogeochemistry for estuaries with different sizes and morphologies along the French Atlantic coast during the period 2014–2016 using readily available geometric, hydraulic, and biogeochemical data. These simulations allowed us to evaluate the budgets of the main nutrients (phosphorus [P], nitrogen [N], silica [Si]) and total organic carbon (TOC), and their imbalance, providing insights into their eutrophication potential. Cumulated average annual fluxes to the Atlantic coast from the seven estuaries studied were 9.6 kt P yr$^{-1}$, 259 kt N yr$^{-1}$, 304 kt Si yr$^{-1}$, and 145 kt C yr$^{-1}$. Retention rates varied depending on the estuarine residence times, ranging from 0–27%, 0–34%, 2–39%, and 8–96% for TP, TN, DSi, and TOC, respectively. Large-scale estuaries had higher retention rates than medium and small estuaries, which we interpreted in terms of estuarine residence times. As shown by the indicator of eutrophication potential (ICEP), there might be a risk of coastal eutrophication, i.e., the development of non-siliceous algae that is potentially harmful to the systems studied due to the excess TN over DSi. This study also demonstrates the ability of our model to be applied with a similar set-up to several estuarine systems characterized by different size, geometries and riverine loads.

**Key words:** estuary; biogeochemical process; retention; C-GEM modelling; French Atlantic coast.

## 1. Introduction

Nutrient transport and transformation along the land–ocean aquatic continuum are receiving increasing attention due to their role in the global nutrient cycle and budget (Howarth et al., 2011, 1991). The estuary is a partly enclosed body of water,

characterized by mixing of salty ocean water with fresh river water (Pritchard, 1967; Vilas et al., 2021), and is an important reactive ecosystem along the land–ocean continuum (Crossland et al., 2005; Regnier et al., 2013). Morphologically, an estuary is an important component connecting land to ocean (Dürr et al., 2011). Freshwater, suspended particulate matter (SPM),

nutrients, and contaminations from watersheds are transferred to the sea and transformed through these interface ecosystems. Ecologically, estuaries are among the most productive ecosystems in the world due to dynamic biogeochemical processes and they ensure many ecological functions that need to be preserved (Barbier et al., 2011; Liquete et al., 2013; Pozdnyakov et al., 2017). They contribute to vegetation growth (phytoplankton, aquatic angiosperms, salt marshes, mangroves, etc.) and animal production (e.g., invertebrates, fish breeding and nursing, bird reproduction and feeding as well as resting areas). Economically,

aquaculture develops around the estuary; agriculture, tourism (port cities), industry and import/export logistics are also active within estuarine basins. Estuarine and coastal ecosystems throughout the world are some of the most heavily used and threatened natural systems (Barbier et al., 2011) despite the efforts to improve natural water quality since the implementation of the Water Framework Directive in the early 2000s (EU Water Framework Directive (WFD), 2000). Although some improvements have been observed in the recent decades regarding some specific perturbations, for instance, a general decrease

in riverine phosphorus loads across Europe, estuaries are still facing significant anthropogenic pressures given that they are the receptacle of all the contaminants and nutrients from the upper river watershed, from both point sources (urban and industrial wastewater) and diffuse sources (agriculture), (Garnier et al., 2021). Additionally, harbors, channelization, and flood protection structures have changed not only the geomorphology of the estuaries, but also their hydrological and biogeochemical behaviors (Romero et al., 2016). These impacts may well be at the heart of critical environmental issues such as estuary and

coastal eutrophication. Eutrophication, i.e., the perturbation of aquatic ecosystems by nutrient enrichment, has been recognized as a serious environmental threat for both continental (lakes and rivers, Vollenweider, 1968) and marine waters (Billen and Garnier, 1997). The amount of nutrients (N, P, Si) delivered to the coastal zone by river systems are often the major determinants of coastal marine eutrophication problems, but previous studies have indicated that eutrophication is not only a result of high inputs of anthropogenic nutrients (N and/or P), but also of their imbalance when anthropogenic N and P are

introduced in excess over Si, the latter resulting from natural rock weathering (Billen and Garnier, 1997; Garnier et al., 2021). The manifestations of eutrophication may take various forms, including harmful algal blooms (Glibert, 2020), which can cause damage to coastal fisheries (Husson et al., 2016). Moreover, overproduction of these algal blooms, which are not suitable for consumption by zooplankton and benthic invertebrates, possibly leads to hypoxia of the bottom water layers (Garnier et al., 2021). Due to the importance of these different aspects of estuaries, their environmental situations and problems are receiving

increasing attention from both researchers and stakeholders.

In order to estimate and possibly control the potential eutrophication or/and hypoxia, the biogeochemical processes (nutrient transport and transformation) in the estuarine systems should be understood. Therefore, accurate quantification of estuarine nutrient fluxes (imports and exports) is necessary for evaluating the "retention" of the system, i.e., the amount of nutrient either sequestered within the estuary and its sediment or eliminated from the system to the atmosphere. In another words, estuarine

retention rates reflect the internal biogeochemical processes and reactions along the estuary, such as uptake, losses,

transformation, changes in storage, mineralization, and degradation. Nutrient retention rates can be mainly influenced by estuarine geomorphology, the surrounding wetlands (most particularly intertidal areas), river discharge, the turbidity maximum zone, estuarine residence time, and other physical forcings (Arndt et al., 2009; Perez et al., 2011). Thus, gaining insight into the retention capacities of estuaries can help understand the biogeochemical process intensities within the estuary and also manage the nutrient imports from the upstream river basins and hence nutrient exports to the seas.

Although the estuarine surface areas are much smaller than those of river networks and coastal marine systems, studying the estuarine ecological function is also complex due to the influences from both the riverine and marine aspects, such as the tide, salinity gradient, estuarine turbidity maximum zone, and hydromorphology (Regnier et al., 1998; Garnier et al., 2008; 2010b; Burchard et al. 2018). To study the estuarine biogeochemical processes, common approaches include in-situ sampling (Coynel et al., 2016; Kaiser et al., 2013; Michel et al., 2000; Modéran et al., 2012; Nguyen et al., 2019; Perez et al., 2011; Savoye et al., 2012) and/or numerical modelling (Arndt et al., 2009; Garnier et al., 2007; Hu and Li, 2009; Laruelle et al., 2019; Ménesguen et al., 2019; Nguyen et al., 2021; Romero et al., 2019). However, direct observations do not allow to quantify nutrient fluxes in macrotidal estuaries because the oscillatory tidal flux is several orders of magnitude larger than the ecologically and biogeochemically-relevant retention flux (Regnier et al., 1998), despite mixing curves are useful for interpreting in situ observations and nutrient dynamics. Moreover, the retention flux generally falls within the range of measurement uncertainties (Arndt et al., 2009; Jay et al., 1997; Regnier et al., 1998). Also, mixing curves are meaningful when water quality data are numerous within the salinity gradient, which is not the case for many estuaries. Numerical models combined with limited observed data can fill the gap in understanding nutrient dynamics and offer insight into past and future scenarios in response to environmental and human changes (such as land use changes and agricultural practices, Billen and Garnier, 2007; Garnier et al., 2021), and climate change (Billen and Garnier, 1997; Garnier et al., 2021). In addition, simulations can be carried out at large spatial and temporal scales. In particular, they can realistically represent the spatial variability within estuaries and provide a global view of the whole basin (land–ocean continuum) by chaining estuarine with river basin models (Laruelle et al., 2019) as well as with coastal zone models (Garnier et al., 2019; Ménesguen et al., 2018b; Romero et al., 2019).

In recent decades, many estuarine numerical models have been applied to disentangle the complex physical and biogeochemical processes, such as 3-D models (Lajaunie-Salla et al., 2017; Romero et al., 2019; Wild-Allen et al., 2013), 2-D models (Arndt et al., 2011; Vanderborght et al., 2007), 1-D models (Hofmann et al., 2008; Volta et al., 2014), and box models (Garnier et al., 2008, 2010b; Verri et al., 2021). However, 3-D models require massive data for calibration and high computing performance to resolve the complex processes occurring in estuaries on relevant spatial and temporal scales. On the opposite side of the complexity spectrum, box models might neglect the transient behavior of the flow and scalar fields and by nature cannot reproduce the complex hydrology of estuarine environments, consequently causing large errors in flux estimations (Arndt et al., 2009). The Carbon-Generic Estuary Model (C-GEM; Volta et al., 2014) used in this study is a depth-averaged 1-D model that has been developed to handle the main obstacles to the application of estuarine models on a regional or global scale (Laruelle et al., 2017). The generic implementation of the C-GEM model relies on a limited amount of basic

information to describe estuarine geometry, hydrodynamic information, and the inputs to estuaries, and then produces annual to multi-decadal simulations. C-GEM has already been applied to one tropical estuary (Nguyen et al., 2021) and several temperate estuaries and has provided satisfactory simulations of nutrient transport and biogeochemical processes despite the simplification of the estuarine geometry (Laruelle et al., 2017, 2019; Volta et al., 2014, 2016a). An extensive description of C-GEM is presented in Volta et al. (2014, 2016a).

The French estuaries along the northeastern Atlantic coast studied herein have been subjected to nutrient enrichment for many years (Garnier et al., 2019; Ménesguen et al., 2019; Ratmaya et al., 2019). Therefore, the objectives of this paper are to: (i) evaluate the nutrient delivery of estuaries from the French Atlantic coast to the sea using the C-GEM modeling approach; (ii) quantify the retention rates of these estuaries; and (iii) analyze the coastal eutrophication potential. The first part of the paper presents longitudinal results averaged over a tidal cycle, including salinity and SPM, nutrients and phytoplankton biomass, under different hydrological years (2014–2016) for seven selected estuaries of different sizes and anthropization levels. Then the annual input–output budgets of nutrients is provided in order to quantify the retention rates of these estuaries according to their specific characteristics (size, degree of hydromorphological management, anthropogenic pressures, etc.).

## 2. Materials and Methods

### 2.1. Study Area

This study focused on seven estuaries (from north to south: Somme, Seine, Vilaine, Loire, Charente, Gironde, and Adour) along the French Atlantic coast (Figure 1). These estuaries embody a wide range of morphological and hydrological settings representative of the region in terms of length, width, residence time, convergence length, tidal amplitude, and length of saline intrusion. Considering these features, they were divided into three large estuaries (the Seine, Loire, and Gironde), two medium-size estuaries (Charente and Adour), and two small ones (Somme and Vilaine). Their geometric properties are presented in Table 1. They are characterized as semi-diurnal and macrotidal estuaries, with an average tidal range from 3.3 m to 5.1 m, and are rather well mixed over the water column (Table 1).

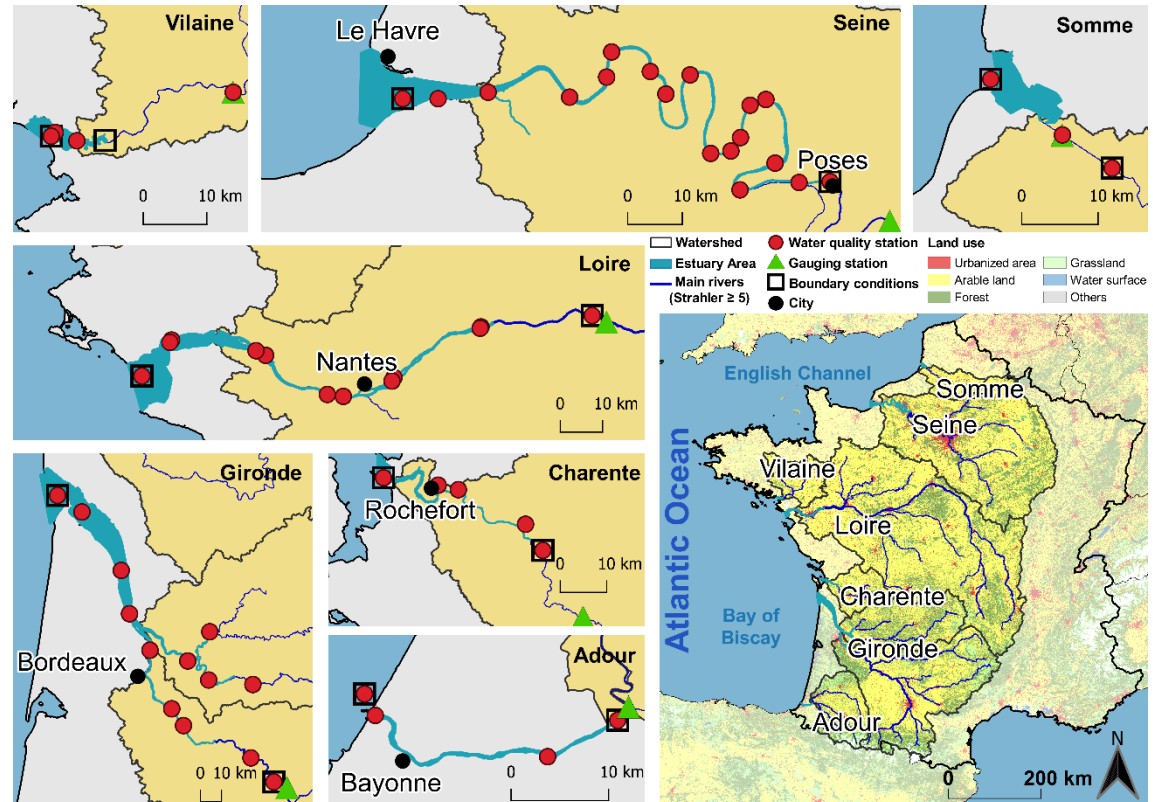

**Figure 1 (a) Map of estuaries studied along the French Atlantic coast. (b) Panels for each estuary indicate the locations of the gauging stations and the water quality stations used in this study.**

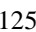125

**Table 1 Geometric properties of the estuaries studied.**

| Estuary | Seine | Loire | Gironde | Charente | Adour | Somme | Vilaine |
|---|---|---|---|---|---|---|---|
| Estuary length (km) | 166 | 122 | 202 | 50 | 34 | 18 | 10 |
| River basin area *(km²) | 65,000 | 111,436 | 75,125 | 7598 | 14,832 | 5560 | 10,498 |
| Estuary basin area (km²) | 11,843 | 6891 | 7677 | 2327 | 2122 | 820 | 237 |
| Depth at estuary mouth (m) | 4.7 | 10.0 | 12.0 | 8.0 | 10.0 | 9.0 | 5.0 |
| Mean tidal range (m) | 5.1 | 4.0 | 3.7 | 4.3 | 3.3 | 3.9 | 3.9 |
| Width at estuary mouth (km) | 10.0 | 10.7 | 12.0 | 1.8 | 0.4 | 3.5 | 2.3 |
| Width at inflection point 1 (km) | 0.48 | 1.5 | 3.6 | 0.3 | - | 0.35 | - |
| Width at inflection point 2 (km) | - | 0.35 | - | - | - | - | - |
| Convergence length (km) | 10.7 | 12.0 | 48.0 | 3.5 | 40.0 | 3.2 | 15.0 |
| Convergence length in the middle of the estuary (km) | - | 21.0 | - | - | - | - | - |
| Convergence length in the upper estuary (km) | 105.5 | 600.0 | 19.0 | 21.0 | - | 4.0 | - |

Note: (1) The geometry information (length, surface area, and width) was derived from remote sensing images through the Esri Ocean layer in Geographic Information System (GIS). Depths were obtained from (Defontaine et al., 2019; Goubert et al., 2019; Laruelle et al., 2019; McLean et al., 2019; Normandin et al., 2019; Toublanc et al., 2015). (2)* River basin areas include the basin area of the rivers flowing directly into the estuary.

These estuaries receive nutrient deliveries from the upper river basins with different land uses and population densities (Table 2): for example, the Seine, Vilaine, Loire, and Charente river basins are dominated by agricultural activities, sustaining a large proportion of the national crop and/or livestock production (Billen and Garnier, 2007; Ménesguen et al., 2018a; Ratmaya et al., 2019). The percentages of different land use types for each basin are presented in Table 2. The intensive agriculture and urbanization within the estuarine basins studied induced eutrophication, especially those of the north half of the French Atlantic coast, regarded as nutrient-enriched (Garnier et al., 2019; Ménesguen et al., 2019; Ratmaya et al., 2019).

**Table 2 Population density (INSEE, 2014) and land use type percentage (CORINE Land Cover, 2018) of the river basin of the systems studied.**

| Basin | Area km$^2$ | Population density inhab km$^{-2}$ | Arable land % | Urban % | Grassland % | Forest % | Water surface % | Others % |
|---|---|---|---|---|---|---|---|---|
| Seine | 65,000 | 256.9 | 66% | 8% | 1% | 25% | 1% | 0% |
| Loire | 111,436 | 74.6 | 72% | 4% | 1% | 21% | 1% | 0% |
| Gironde | 51,343 75125 | 95.6 | 56% | 3% | 6% | 33% | 1% | 1% |
| Charente | 7598 | 79.0 | 77% | 5% | 1% | 18% | 0% | 0% |
| Adour | 14,832 | 72.8 | 48% | 4% | 16% | 27% | 0% | 4% |
| Somme | 5560 | 115.1 | 84% | 7% | 0% | 8% | 1% | 1% |
| Vilaine | 10,498 | 106.7 | 82% | 6% | 1% | 11% | 0% | 0% |

## 2.2. Data Collection

The measured data used in this study to calibrate and validate the model, as well as to determine the upstream limit conditions, include river discharge (Q), salinity (Sal), suspended particulate matter (SPM), and water quality variables, i.e., phosphate (PO$_4$), ammonium (NH$_4$), nitrate (NO$_3$), dissolved silica (DSi), dissolved oxygen (DO), total (particulate and dissolved) organic carbon (TOC), and chlorophyll a (Chl-a), as an indicator of phytoplankton biomass.

Daily river discharge data were obtained from the national Banque Hydro database (http://www.hydro.eaufrance.fr/). Water quality variables were collected from the national database: (1) the French Water Agencies through the NAIADES portal (http://naiades.eaufrance.fr/) for the physicochemical parameters of surface waters; (2) the REPHY (REPHY, 2021) database (https://www.seanoe.org/data/00361/47248/) specifically for the monitoring of phytoplankton for the estuarine and marine parts; and (3) the SOMLIT database (https://www.somlit.fr/) for coastal and estuarine water quality parameters. The temporal resolution of the water quality data acquired is generally monthly or bimonthly. Chl-a concentrations were usually available only from March to September (main phytoplankton growth period). The gauging stations and water quality stations are located in Figure 1(b).

The data for the marine boundaries were extracted from outputs of simulations performed by a coastal marine model ECO-MARS3D (Cugier et al., 2005b; Lazure and Dumas, 2008; Ménesguen et al., 2019), providing water elevation, temperature, salinity, SPM, PO$_4$, NH$_4$, NO$_3$, DSi, DO, TOC, and phytoplankton (Phy) at 4-km spatial and hourly temporal resolutions.

The use of the above database is detailed in section 2.3.3.

## 2.3. Modelling Approach and Setting

### 2.3.1.   C-GEM

In this study, the C-GEM (Carbon Generic Estuarine Model) was used for modeling nutrient transport and transformation for the selected estuaries. The C-GEM is a depth-averaged 1-D process-based model designed to simulate estuarine hydrodynamics, transport, and the biogeochemistry of tidal alluvial estuaries with relatively little data and computation demand (Volta et al., 2014, 2016b). A 1-D model can be considered as well adapted to these shallow macrotidal estuaries that mixed at each tide cycle as shown by Brion et al. (2000) on the Seine River and Middelburg and Herman (2007) for other estuaries of the Atlantic Coast of Europe, which was also supported by Savenije (2012). The extensive description of the model and its underlying assumptions are available in Volta et al. (2014). Thus, the following sections only briefly describe the state variables and processes included in the biogeochemical module as well as the modifications introduced for the simulations discussed in this manuscript. Furthermore, all the equations governing the production and consumption reactions of all state variables as well as their parameterization are provided in the supplementary material (Table S-1).

### 2.3.2.   Model Description and Set-up

C-GEM is based on the premise that geometry and hydrodynamics exert first-order control on the estuarine transport and biogeochemical processes (Volta et al., 2014). It uses idealized geometry (defined by the estuarine width at the mouth, convergence length, channel depth profile; see Volta et al., 2014 for details) and hydrodynamics (such as river discharge and tidal amplitude) that can be gained from remote sensing images using Geographic Information Software (GIS) and readily available data sets (Table 1).

The biogeochemical reaction network includes SPM settling and erosion, the air–water gas exchange for oxygen ($O_2$) and carbon dioxide ($CO_2$), nitrification, denitrification, primary production, phytoplankton mortality, and aerobic degradation of organic matter (see supplementary material (SM) according to Volta et al. (2014a, 2016a) for detailed descriptions and mathematical formulations). Essential state variables are used in the simulations, as are those gathered above (mentioned in section 2.2: DO, $NH_4$, $NO_3$, $PO_4$, DSi, TOC, and diatom (Dia) and non-diatom (nDia)) pools are therefore considered in this study and schematized in Figure 2. Note that, in this study, the inorganic carbon module of C-GEM (which includes the explicit calculation of dissolved inorganic carbon, alkalinity, $pCO_2$, and pH) described in Volta et al. (2014) was not activated. The SPM dynamics is controlled by transport of suspended material (i.e., advection and dispersion) as well as local deposition and resuspension/erosion processes, but the model does not distinguish between the pools of marine and riverine suspended material. P adsorption and desorption to particulate material to form an iron-bound complex for example, is not accounted for. Thus, the only control exerted by SPM concentrations in the water column on the other biogeochemical variables occurs through the influence of SPM on the light extinction coefficient, which partly controls primary production. While C-GEM

does not yet include an explicit benthic compartment, a net burial term was applied to the particulate state variables of the model, namely: Dia, nDia, and TOC. This term provides a first-order representation of the permanent removal of particulate material through sediment accumulation. It is applied to phytoplankton and TOC, proportionally to their concentration and inversely proportionally to the depth of the water column, using a constant settling rate of 1 m d$^{-1}$ for phytoplankton and 0.4 m d$^{-1}$ for TOC. Note that the concentrations of the organic state variables (Dia, nDia and TOC) are expressed in µmol of carbon per liter but the model uses the Redfield ratios to account for the associated amounts of N, P and, in the case of diatoms, of Si. Thus, the variable TOC actually includes all detritus and is sustained by the death of phytoplankton and its aerobic degradation fuels the stocks of dissolved inorganic nutrients.

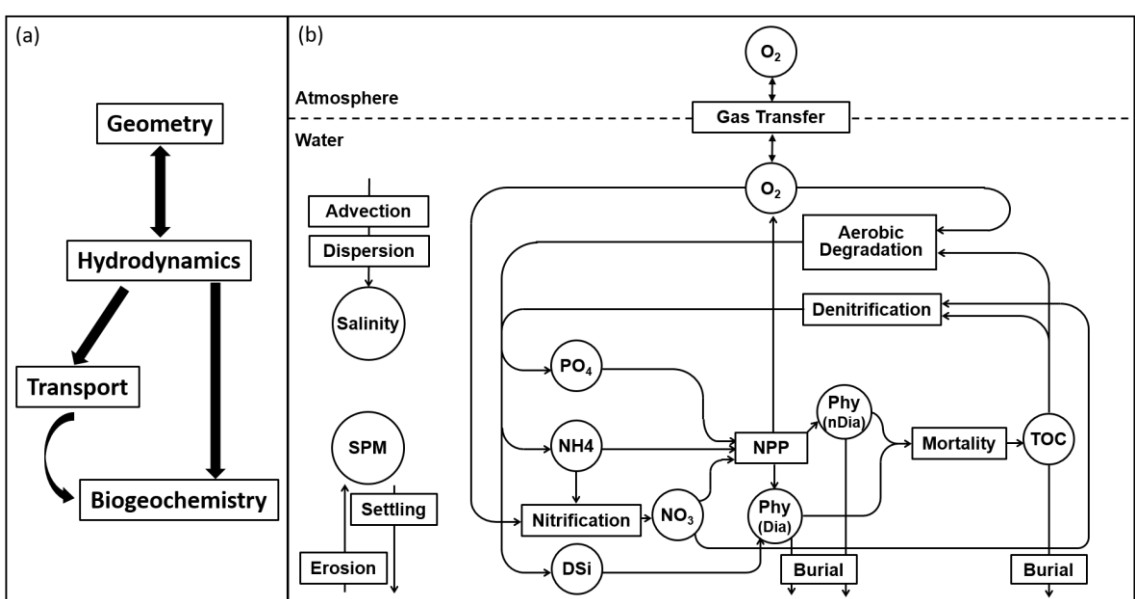

**Figure 2 (a) The C-GEM concept. (b) Conceptual scheme of the biogeochemical module used in C-GEM in this study: a circle represents the state variables while a rectangle represents the processes; Dia corresponds to diatoms.**

### 2.3.3.    Forcings and Boundary Conditions

C-GEM is constrained by a set of riverine and marine boundary conditions. The riverine boundary conditions include river discharge (Q), SPM, and the state variables of water quality concentrations, which were linearly interpolated to obtain the daily values between the adjacent available measurements (Table 3). The marine boundary conditions include water elevation, water temperature, salinity, and the same water quality variables extracted from ECO-MARS3D at an hourly temporal resolution that allows capturing the tidal cycle. SPMs for the marine boundary conditions were collected from REPHY (REPHY, 2021) and linearly interpolated to obtain the data at the required time scale (Table 3). The annual means of boundary conditions are summarized in Table 4.

**Table 3 State variables in C-GEM and the boundary conditions.**

| State variable (abbreviation) | Unit in | Marine boundary condition | | Riverine boundary condition | |
|---|---|---|---|---|---|
| | | Source | Timestep | Source | Timestep |

| | C-GEM | | | | | |
|---|---|---|---|---|---|---|
| Discharge (Q) | $m^3/s$ | - | - | NAIADES | daily | |
| Water elevation | m | ECO-MARS3D | hourly | - | - | |
| Water temperature | °C | ECO-MARS3D | hourly | - | - | |
| Salinity (Sal) | - | ECO-MARS3D | hourly | NAIADES | daily | |
| Suspended particulate matter (SPM) | g L$^{-1}$ | REPHY | hourly | NAIADES | daily | |
| Phosphate (PO$_4$) | µmol P L$^{-1}$ | ECO-MARS3D | hourly | NAIADES | daily | |
| Ammonium (NH$_4$) | µmol N L$^{-1}$ | ECO-MARS3D | hourly | NAIADES | daily | |
| Nitrate (NO$_3$) | µmol N L$^{-1}$ | ECO-MARS3D | hourly | NAIADES | daily | |
| Dissolved silica (DSi) | µmol Si L$^{-1}$ | ECO-MARS3D | hourly | NAIADES | daily | |
| Total organic carbon (TOC) | µmol C L$^{-1}$ | ECO-MARS3D | hourly | NAIADES | daily | |
| Dissolved oxygen (DO) | µmol O$_2$ L$^{-1}$ | ECO-MARS3D | hourly | NAIADES | daily | |
| Phytoplankton (Phy) | µg Chl L$^{-1}$ | ECO-MARS3D | hourly | NAIADES | daily | |

Note: Calculations by C-GEM are in µmole for C, N, P, Si, O$_2$, but output values of the model are provided in mass of these elements (by multiplying the µmole L$^{-1}$ by respectively 12, 14, 31, 28 and 32 for a unit in µg (C, N P, Si, O$_2$) L$^{-1}$.

In this study, the Dordogne River, a tributary of the Gironde estuary, which contributes ~40% discharge to the estuary, was considered as an upstream condition. The closest available sampling station (number: 5026000, from NAIADES) on the Dordogne river is around 32 km upstream from the confluence, in a stretch influenced by tide. The inputs from the Dordogne tributary include daily Q and the same water quality variables (listed in Table 3) for riverine boundary conditions. Intermittent observed water quality data were also interpolated to daily values.

### 2.3.4. Point Sources

The discharge from waste water treatment plants (WWTPs) was taken into account as point sources in the model. The WWTPs from the largest estuarine cities (above 50,000 inhabitants) were considered: Rouen, Nantes, Rochefort, Bordeaux, and Bayonne on the Seine, Loire, Charente, Gironde, and Adour rivers, respectively. Fluxes from WWTPs to estuaries were calculated based on the loads treated in each WWTP (expressed in inhabitant equivalents), the identification of treatment types and assuming specific per capita emissions (after appropriate treatment) of SPM, PO$_4$, NH$_4$, NO$_3$, DSi, TOC, DO (state variables of the model), with an average water release of 150 L inhab$^{-1}$ day$^{-1}$ (Table 4).

**Table 4 Annual mean of inputs (riverine boundary conditions) and point source (waste water treatment plants (WWTPs)) for the estuaries studied. The standard deviations provided do not correspond to uncertainties but express the seasonal variability of the concentrations at the boundary conditions. Time-series of those concentrations are provided in the supplementary table S-4.**

| Estuary | Q m$^3$ s$^{-1}$ | Sal psu | SPM mg L$^{-1}$ | PO$_4$ mgP L$^{-1}$ | NH$_4$ mgN L$^{-1}$ | NO$_3$ mgN L$^{-1}$ | DSi mgSi L$^{-1}$ | TOC mgC L$^{-1}$ | DO mgO$_2$ L$^{-1}$ | Chl-a ug L$^{-1}$ |
|---|---|---|---|---|---|---|---|---|---|---|
| **Riverine boundary** | | | | | | | | | | |
| Seine | 499.4±282.5 | - | 21.5±20.2 | 0.102±0.034 | 0.149±0.276 | 4.9±0.8 | 3.7±0.7 | 3.5±0.7 | 9.4±1.6 | 2.1±1.2 |
| Loire | 808.1±619.3 | - | 19.4±11.9 | 0.034±0.019 | 0.017±0.011 | 3.0±0.5 | 5.3±1.1 | 5.6±1.9 | 10.2±1.5 | 9.3±7.6 |
| Gironde (without Dordogne) | 521±386.7 | - | 19.6±26.5 | 0.034±0.013 | 0.052±0.036 | 1.6±0.6 | 2.9±0.9 | 2.5±0.7 | 9.8±1.5 | 1.8±3.1 |
| *Dordogne** | *303.4±261.7* | - | *827.0±1460.4* | *0.043±0.020* | *0.047±0.092* | *1.5±0.3* | *4.4±0.8* | *3.2±1.0* | *9.2±2.2* | *2.8±3.2* |
| Charente | 77.3±85.1 | - | 9.5±8.5 | 0.036±0.012 | 0.052±0.023 | 5.4±1.0 | 5.2±1.3 | 2.6±0.8 | 9.1±1.4 | 1.4±0.4 |
| Adour | 324.9±293.2 | - | 20.9±26.9 | 0.039±0.018 | 0.080±0.041 | 1.6±0.5 | 2.7±0.3 | 2.5±0.9 | 9.4±1.2 | 1.2±0.4 |
| Somme | 48.2±7.5 | - | 9.0±6.8 | 0.033±0.016 | 0.063±0.024 | 4.4±0.6 | 8.4±1.2 | 2.6±0.3 | 9.8±1.6 | 2.1±2.3 |
| Vilaine | 95.6±140.0 | - | 12.6±6.7 | 0.034±0.018 | 0.045±0.026 | 4.6±2.0 | 6.7±0.9 | 3.5±1.4 | 10.3±1.4 | 14.5±14.8 |
| **Marine boundary** | | | | | | | | | | |
| Seine | - | 15.1±3.3 | 3.6±2.5 | 0.071±0.020 | 0.027±0.027 | 2.8±0.5 | 1.9±0.4 | 0.7±0.1 | 9.6±1.1 | 6.7±1.5 |
| Loire | - | 23.1±2.4 | 43.1±37.6 | 0.039±0.008 | 0.015±0.007 | 1.0±0.4 | 1.9±0.4 | 0.7±0.3 | 7.3±1.5 | 1.9±1.4 |
| Gironde | - | 30.1±2.1 | 36.5±20.2 | 0.021±0.006 | 0.010±0.005 | 0.4±0.3 | 0.6±0.2 | 0.1±0.1 | 7.8±1.2 | 1.2±1.3 |

| | | | | | | | | | | |
|---|---|---|---|---|---|---|---|---|---|---|
| Charente | - | 22.8±2.3 | 23.3±19.2 | 0.032±0.007 | 0.017±0.008 | 2.1±0.4 | 1.9±0.3 | 0.6±0.1 | 8.4±0.9 | 1.4±1.0 |
| Adour | - | 16.2±0.2 | 7.8±13.3 | 0.041±0.003 | 0.016±0.003 | 0.8±0.0 | 1.5±0.0 | 0.7±0.1 | 7.3±1.0 | 0.6±0.7 |
| Somme | - | 28.7±2.2 | 21.5±24.3 | 0.014±0.005 | 0.018±0.009 | 0.9±0.3 | 1.1±0.3 | 0.4±0.2 | 9.0±0.9 | 4.7±1.2 |
| Vilaine | - | 23.2±3.2 | 17.2±14.4 | 0.024±0.011 | 0.045±0.028 | 1.9±0.7 | 1.5±0.4 | 1.9±0.3 | 8.7±1.0 | 4.5±1.6 |
| **Point source\*\*** | | | | | | | | | | |
| Rouen WWTP | 0.88 | - | 26.7 | 0.700 | 6.667 | 16.7 | 2 | 30.9 | 4.1 | - |
| Nantes WWTP | 1.35 | - | 26.7 | 0.700 | 6.667 | 16.7 | 2 | 30.9 | 4.1 | - |
| Rochefort WWTP | 0.11 | - | 26.7 | 0.700 | 6.667 | 16.7 | 2 | 30.9 | 4.1 | - |
| Bordeaux WWTP | 2.04 | - | 39.9 | 2.903 | 8.880 | 20 | 2 | 35.1 | 4.1 | - |
| Bayonne WWTP | 0.29 | - | 26.7 | 0.700 | 6.667 | 16.7 | 2 | 30.9 | 4.1 | - |

* The Dordogne River, which joins the Gironde estuary, is considered as an input to the estuary, in a way similar to the point sources, but with a daily time step (see Table 3), contrary to point sources** from WWTP inputs to the estuary for which we considered an annual average.

### 2.3.5. Simulation Set-Up

C-GEM simulates the transport and transformation in sequence by applying an operator splitting approach (Volta et al., 2014) and a finite difference scheme of a regular grid at 2 km with a time step of 300 s. The simulation starts following a 60-day spin-up during which only the hydrodynamics and transport modules are resolved over a repeating identical tidal cycle, which enables the system to reach a dynamic steady-state, providing realistic initial conditions for the biogeochemical module. The relatively short residence time, combined with hourly boundary conditions at the mouth of the estuary allows an accurate resolution of the tidal cycle in the estuaries, including during transient simulations as was demonstrated by Laruelle et al. (2019). The time resolution of the model outputs was set at 4 hours in order to minimize the size of the export files while capturing most of the tidal and diurnal variability. In the figures there-after, the envelope around the model results represents the minimum and maximum values over two tidal cycles in order to provide the amplitude of the tidal influence on concentrations at different locations of the estuary. In this study, the model was run over a 3-year period from 2014 to 2016 for each estuary.

### 2.4. Model Calibration

The data used to calibrate and validate the model were obtained from the multiple databases mentioned in section 2.2. The locations of these stations along the estuaries are presented in Figure 1. Calibration was implemented based on 2015, an intermediate annual flow between those of wet and dry years. The performance of the hydrodynamics modules was first evaluated by comparing the simulated salinity with observed salinity along the estuary. Then the mass transport was calibrated using longitudinal SPM profiles as an indicator. Good agreement with in situ data for salinity and SPM was reached by setting a maximum value of 100 m s$^{-2}$ to the dispersion coefficient generated by the equations described in Volta et al. (2014). This dispersion value is in agreement with the previous study in the Scheldt estuary (Arndt et al., 2009) and the Loire estuary (Thouvenin et al., 1997). The values used for all parameters involved in the hydrodynamics and transport modules are presented in Table 5.

**Table 5 Parameter settings related to salinity and sediment and their values along the estuaries (low, middle, and upstream parts).**

| Estuary | Settling velocity ($m\ s^{-1}$) | Chezy coefficient ($m^{0.5}\ s^{-1}$) | | | Erosion coefficient ($mg\ m^{-2}\ s^{-1}$) | | Critical shear stress ($N\ m^{-2}$) | |
|---|---|---|---|---|---|---|---|---|
| | | Low | Middle | Up | Low | Up | Low | Up |
| Seine | 0.001 | 85 | 70 | 40 | $1.6*10^{-5}$ | $1.3*10^{-6}$ | 0.35 | 1.20 |
| Loire | 0.001 | 50 | - | 20 | $5.0*10^{-5}$ | $1.0*10^{-6}$ | 0.30 | 1.00 |
| Gironde | 0.001 | 80 | 40 | 35 | $2.0*10^{-4}$ | $8.8*10^{-6}$ | 0.30 | 1.00 |
| Charente | 0.001 | 50 | - | 20 | $9.5*10^{-5}$ | $8.0*10^{-6}$ | 0.25 | 1.00 |
| Adour | 0.001 | 40 | - | 20 | $8.0*10^{-5}$ | $5.0*10^{-6}$ | 0.30 | 0.80 |
| Somme | 0.001 | 50 | - | 30 | $9.0*10^{-5}$ | $2.0*10^{-7}$ | 0.35 | 1.20 |
| Vilaine | 0.001 | 30 | - | 20 | $9.0*10^{-5}$ | $8.0*10^{-6}$ | 0.30 | 1.20 |

Once the hydrodynamics was satisfactorily calibrated (see section 3.1 ), the biogeochemical module was calibrated using the observed water quality data (2015), starting from the parameterization described in Laruelle et al. (2019a), which was used for an application of C-GEM to the Seine estuary. The same set of values for all biogeochemical parameters was used for all of the selected estuaries (Table 6). On the whole, only slight variations from the parameterization in Laruelle et al. (2019a) were required, ensuring that all parameters remained within a range corresponding to values representative of temperate estuaries following the extensive literature survey carried out by Volta et al. (2016a).

**Table 6 Biogeochemical parameter values used in C-GEM. See the formulations of the processes in the supplementary material.**

| Parameter name in C-GEM | Description | Unit | Value | Reference range (Volta et al., 2016a) |
|---|---|---|---|---|
| $P^B_{max}[Phy]$ | Maximum specific photosynthetic rate | $s^{-1}$ | $7.0*10^{-5}$ | $1.07*10^{-6} \sim 1.82*10^{-4}$ |
| $\alpha[Phy]$ | Photosynthetic efficiency | $m^2\ s\ (\mu E*s)^{-1}$ | $5.0*10^{-7}$ | $1.67*10^{-7} \sim 6.94*10^{-7}$ |
| kmort [Phy] | Phytoplankton mortality rate constant | $s^{-1}$ | $3.85*10^{-7}$ | $2.3*10^{-7} \sim 2.35*10^{-5}$ |
| kgrowth[Phy] | Phytoplankton growth constant | - | 0.29 | $0.1 \sim 0.5$ |
| kmaint[Phy] | Phytoplankton maintenance rate constant | $s^{-1}$ | $4.6*10^{-7}$ | $1.6*10^{-7} \sim 3.5*10^{-6}$ |
| kexcr[Phy] | Excretion constant | - | 0.05 | $0.03 \sim 0.07$ |
| vPHY | Settling velocity of phytoplankton | $m\ s^{-1}$ | $1.16*10^{-5}$ | - |
| vTOC | Settling velocity of organic matter | $m\ s^{-1}$ | $4.6*10^{-6}$ | - |
| $K_{D1}$ | Background light attenuation | $m^{-1}$ | 0.3 | - |
| $K_{D2}$ | SPM light attenuation | $(mg*m)^{-1}$ | 0.03 | - |
| $KPO_4$ | Michaelis–Menten constant for phosphate | $\mu mol\ P$ | 0.2 | $0.001 \sim 3.58$ |
| KN | Michaelis–Menten constant for dissolved nitrogen | $\mu mol\ N$ | 1.13 | $0.1 \sim 7.14$ |
| $KNH_4$ | Michaelis–Menten constant for ammonium | $\mu mol\ N$ | 80 | $71.43 \sim 643.0$ |
| $KNO_3$ | Michaelis–Menten constant for nitrate | $\mu mol\ N$ | 26.07 | $7.14 \sim 45.0$ |
| KDSi | Michaelis–Menten constant for dissolved silica | $\mu mol\ Si$ | 1.07 | $0.3 \sim 20.0$ |
| KTOC | Michaelis–Menten constant for organic carbon | $\mu mol\ C$ | 300 | $60 \sim 312.5$ |
| $KO_2\_ox$ | Michaelis–Menten constant for aerobic degradation | $\mu mol\ O_2$ | 31 | $15 \sim 34$ |
| $KO_2\_nit$ | Michaelis–Menten constant for nitrification | $\mu mol\ O_2$ | 51.25 | $15 \sim 312.5$ |
| $KinO_2$ | Inhibition constant for nitrification | $\mu mol\ O_2$ | 33 | $15 \sim 63$ |
| Kox | Aerobic degradation rate constant | $\mu mol\ C\ s^{-1}$ | $2.0*10^{-4}$ | $9.75*10^{-5} \sim 9.26*10^{-4}$ |
| Knit | Nitrification rate constant | $\mu mol\ N\ s^{-1}$ | $5.0*10^{-4}$ | $1.06*10^{-5} \sim 2.17*10^{-3}$ |
| Kdenit | Denitrification rate constant | $\mu mol\ N\ s^{-1}$ | $1.0*10^{-3}$ | $2.6*10^{-5} \sim 5.22*10^{-1}$ |
| redsi | Redfield ratio for silica | mol Si/mol C | 15/106 | - |
| redn | Redfield ratio for nitrogen | mol N/mol C | 16/106 | - |
| redp | Redfield ratio for phosphorus | mol P/mol C | 1/106 | - |

After calibration of the hydrodynamic and biogeochemical processes for 2015, the model was then also implemented and run for 2014 and 2016. The model was validated for 2014 and 2016, using the parameterization constrained by the calibration over 2015. The results were evaluated against field measurements using widely used statistical indicators, namely, the root mean

squared error (RMSE) and bias (Moriasi et al., 2007), which were calculated as:

$$\text{RMSE} = \sqrt{\sum_{i=1}^{n}(obs_i - sim_i)^2 / n}$$

$$\text{Bias} = \sum_{i=1}^{n}(obs_i - sim_i) / \sum_{i=1}^{n} obs_i$$

where $n$ is the number of samples, $obs$ is the observation, and $sim$ is the simulation. Evaluations are shown in section 3.1.

## 2.5. Calculation of Indicators of Estuarine Ecological Functions

To perform a year-long calculation of fluxes entering and leaving the estuarine system, the quantities transported through the boundaries of the system for each state variable were computed at each time step by the advection and dispersion schemes and integrated over the time period considered. This calculation and the relatively short computation time step of 150 s used by the model ensured accurate representation of the transient processes taking place during the tidal cycle in the estuary.

In this study, dissolved inorganic nitrogen (DIN, considered as the sum of $NH_4$ and $NO_3$); dissolved inorganic phosphorus (DIP equals $PO_4$); dissolved silica (DSi), Phy (Chl-a, using a C/Chl-a ratio of 40 (Jakobsen and Markager, 2016)) and TOC were calculated (Table 8). In addition, the Redfield–Brzezinkski ratios C:N:P:Si = 106:16:1:15–20 (Brzezinski, 1985; Redfield et al., 1963) were used to take into account the organic fractions and estimate total nitrogen (TN), total phosphorus (TP), and total silica (TSi), whose fluxes were preferentially chosen for calculating overall retention rates.

The retention rate of the estuary represents the intensity of the nutrient retention and/or elimination and/or transformations within the estuary and is calculated as:

$$\text{Retention Rate} = (\text{Flux}_{imp} - \text{Flux}_{exp}) / \text{Flux}_{imp} \times 100$$

where $Flux_{imp}$ is the annual sum of all the imports to any of the estuaries for DIP, DIN, DSi, TOC; $Flux_{exp}$ are the annual exports at the outlet of the estuaries.

The estuarine residence time was calculated considering the estuarine geomorphology and the river discharge to the estuary and specifically represents the fresh water residence time;

$$\text{Residence Time} = V/Q$$

where $V$ is the estuarine volume ($m^3$) calculated by the integral from the estuarine length, width, and mean depth; $Q$ ($m^3 s^{-1}$) is the river mean annual discharge entering the estuary.

The indicator for coastal eutrophication potential (ICEP, Billen and Garnier, 2007; Garnier et al., 2010a) allows quantifying estuarine nutrient flux balance or imbalance entering the coastal zone considering the excess of nitrogen (N) and phosphorus (P) to silica (Si), taking into account the N:P:Si stoichiometry according to the algae requirement. The ICEP is calculated as:

$$\text{P-ICEP} = [\text{Flux P}/31 - \text{Flux Si}/28 \times \text{P/Si}] \times \text{C/P} \times 12$$

$$\text{N-ICEP} = [\text{Flux N}/14 - \text{Flux Si}/28) \times \text{N/Si}] \times \text{C/N} \times 12$$

The P-ICEP and N-ICEP are expressed in carbon flux units (kg C $km^{-2}$ $d^{-1}$), using the C:N and C:P Redfield ratios (Brzezinski, 1985; Redfield et al., 1963). Flux P, Flux N, and Flux Si are the mean specific fluxes of total phosphorus, total nitrogen, and dissolved silica, respectively, delivered at the outlet of the estuary, expressed in kg P $km^{-2}$ $d^{-1}$, kg N $km^{-2}$ $d^{-1}$, and kg Si $km^{-2}$

$d^{-1}$.

### 3. Results

### 3.1. Model Performance

#### 3.1.1.    Calibration Based on 2015

The model was calibrated using available observations for the year 2015. The results from simulations performed using C-
GEM and the corresponding observations were plotted for two dates in 2015 (one in winter [January] and the other in summer
[July]) along the estuarine length for salinity, SPM, and water quality variables ($PO_4$, $NH_4$, $NO_3$, DSi, TOC, DO, Chl-a; Figure
3 and Figure S-1 in SM). The model was run using forcings and boundary conditions representative of the sampling dates of
the field data.

The hydrodynamics module was calibrated on its ability to reproduce the salinity intrusion into the estuary under different
conditions. Simulations followed the same increasing tendency with the measurements. In January, with high river discharges,
the salinity intrusion reached 25 km from the sea mouth for large estuaries, such as the Seine and Loire estuaries, and within
50 km for the largest, the Gironde estuary. For the medium-size estuaries such as the Charente and Adour, salinity decreased
to 1 at ~15 km from the sea mouth. During low discharges in July, salinity penetrated farther into the inner estuary, up to ~40–
60 km from the sea mouth for large estuaries, while for the medium-size estuary it reached ~20 km from the sea mouth. Such
seasonal and scale-dependent patterns for salinity agree closely with observations.

Regarding SPM calibration, the model better captured the dynamics for the large estuaries than for smaller ones (Figure 3 and
Figure S-1). Simulated SPM values were in the range of measured peaks, although some observations may be missing for the
medium-size and small estuaries. The model generally provided an adequate representation of the turbidity maximum zone
(TMZ). The Seine showed characteristics similar to the Loire, as they both showed a maximum SPM concentration at ~250
315    mg $L^{-1}$ with TMZ quite close to the sea mouth (~20–30 km) in high discharge season and a maximum SPM concentration at
~500 mg $L^{-1}$ with a TMZ up to ~40–50 km in low discharge season. The SPM of the Gironde ranged from ~1000 to ~2000 mg
$L^{-1}$ for the selected dates in 2015, with a TMZ located more upstream to the river, ~90–130 km to the sea mouth. Simulated
SPM was low for the Charente, from ~70 mg $L^{-1}$ in July to ~120 mg $L^{-1}$ in January, and less than 10 mg $L^{-1}$ for the Adour for
both dates, but observations were very scarce.

Simulations of longitudinal variations in concentration for DIP, $NH_4$, $NO_3$, DSi, and TOC for the two dates selected matched
available observations. All nutrients, organic carbon, and Chl-a concentrations decreased mostly at the mouth, due to seawater
dilution. In winter, high discharge led to increasing transport of elements (more than transformation), and longitudinal
variations appeared rather flat. In summer, longitudinal profiles showed more transformation, before dilution at sea;
development of phytoplankton also occurred within the estuaries, in the fluvial or saline sections (Figure 3).

$PO_4$ concentrations mostly varied within 0.1 mg P $L^{-1}$ (3.2 µmol $L^{-1}$) except for the Seine whose maximum concentration can

reach 0.2 mg P L$^{-1}$ (6.5 µmol L$^{-1}$) in July with low discharge. Most observations were captured by the model, though at some stations the simulations overestimated the observations. NH$_4$ concentrations for the selected dates never exceeded 0.1 mg N L$^{-1}$ (7.1 µmol L$^{-1}$) except for the Adour at the first 15 km from the riverine boundary, with values varying from 0.1 to 0.15 mg N L$^{-1}$ (7.1 to 10.7 µmol L$^{-1}$). NO$_3$ concentrations of the Seine, Loire, and Charente were higher (4–5 mg N L$^{-1}$, e.g., 285.7-

357.1 µmol L$^{-1}$) than the ones of the Gironde and Adour (~1.5–2.0 mg N L$^{-1}$, e.g., ~107.1-142.9 µmol L$^{-1}$). Simulated NO$_3$ underestimated the observations for the Loire in winter (in January at the ~ 30~70 km from the sea mouth), and slightly overestimated the NO$_3$ values for the Gironde at ~40–100 km from the sea mouth downstream of the confluence with the Dordogne tributary. DSi concentrations showed the highest concentrations for the Loire (7.0 mg Si L$^{-1}$, e.g., 250.0 µmol L$^{-1}$), the lowest for the Charente and Adour (~2 mg Si L$^{-1}$, e.g., 71.4 µmol L$^{-1}$), and intermediate for the Seine and Gironde (~5 mg

Si L$^{-1}$, e.g., 178.6 µmol L$^{-1}$). Regarding the TOC levels, they were on the order of 5 mg C L$^{-1}$ (416.7 µmol L$^{-1}$) for these five estuaries: slightly lower in summer, except for the Seine (Figure 3).

For each variable, the fit between simulations and observations for the calibration year (2015) was evaluated through the value of Bias and RMSE (Table 7). The model was also evaluated for the validation years (2014 and 2016, Table 7).

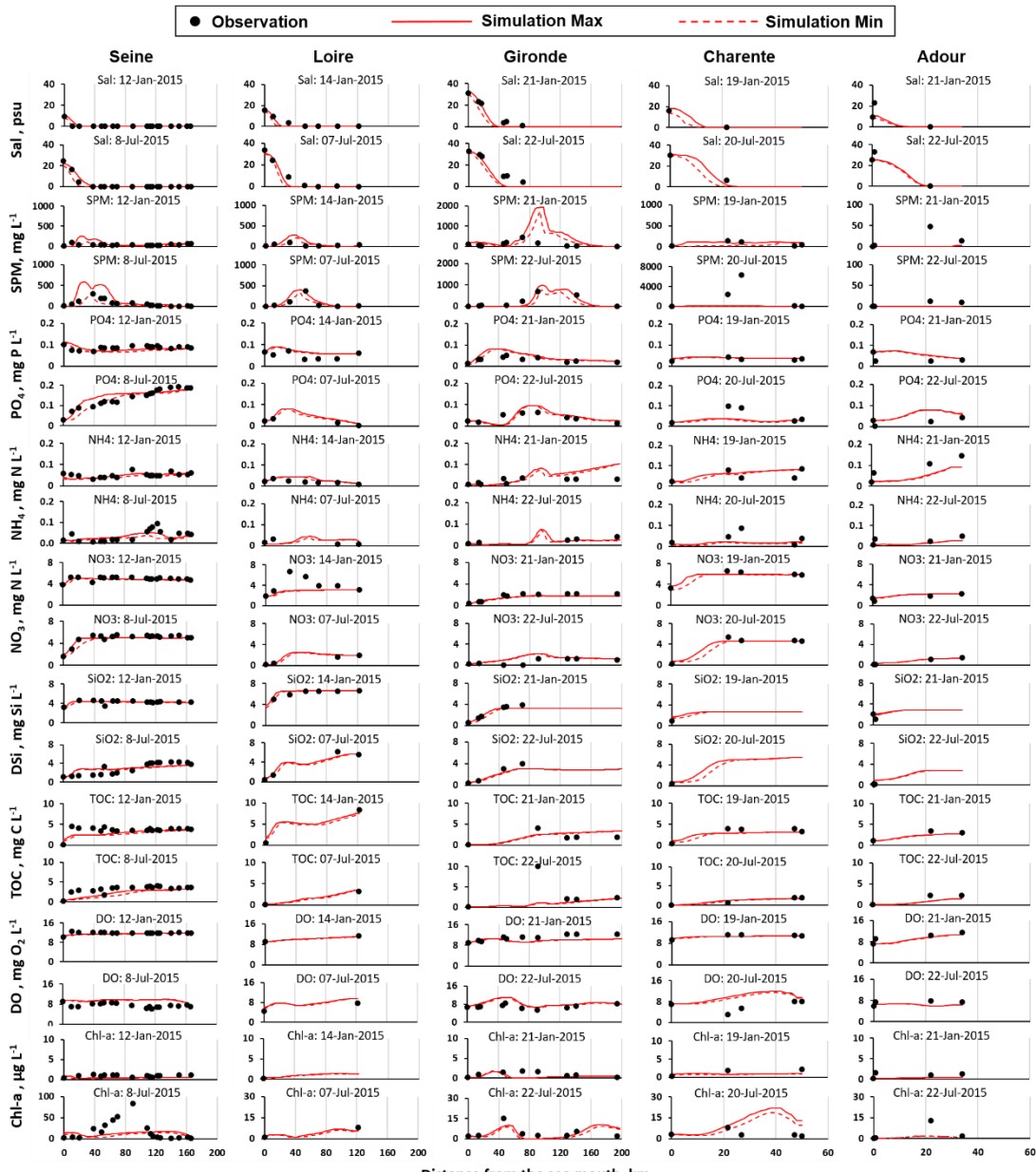

**Figure 3 Salinity (Sal), suspended particulate matter (SPM), and nutrients (PO4, NH4, NO3, DSi, TOC), dissolved oxygen (DO), chlorophyll a (Chl-a) concentration variations along the estuaries (the Seine, Loire, Gironde, Charente, and Adour) for two selected dates (one in winter and the other one in summer). Note the different scales for the SPM and Chl-a for the estuaries. The results of the Somme and Vilaine estuaries are shown in the supplementary material.**

The model outputs were compared to available observations and provided statistical indicators (bias and RMSE , Table 7) which reflect overall good performances of the model ($-0.7 < $ Bias $ < 0.7$) through the standards provided by Moriasi et al. (2007).

The RMSEs in the calibration period remained within the range of standard deviation values provided in Table 4 for upstream boundary conditions.

**Table 7 Bias and root mean square error (RMSE) for salinity (Sal), phosphate (PO$_4$), ammonia (NH$_4$), nitrate (NO$_3$), dissolved silica (DSi), total organic carbon (TOC) for the Seine, Loire, Gironde, Charente, Adour, Somme, and Vilaine estuaries.**

| Variables | Unit | Estuary | Calibration | | | Validation | | |
|---|---|---|---|---|---|---|---|---|
| | | | n | Bias | RMSE | n | Bias | RMSE |
| Sal | psu | Seine | 132 | −0.29 | 0.68 | 266 | −0.17 | 1.34 |
| | | Loire | 60 | 0.30 | 4.83 | 120 | 0.39 | 4.05 |
| | | Gironde | 34 | 0.41 | 5.62 | 62 | 0.40 | 5.11 |
| | | Charente | 0 | - | - | 2 | NR* | NR* |
| | | Adour | 12 | 0.34 | 11.05 | 26 | 0.21 | 8.86 |
| | | Somme | 24 | 0.11 | 2.79 | 51 | 0.11 | 3.44 |
| | | Vilaine | 24 | 0.15 | 6.87 | 48 | 0.12 | 7.04 |
| PO$_4$ | mgP L$^{-1}$ | Seine | 150 | 0.01 | 0.02 | 299 | −0.03 | 0.02 |
| | | Loire | 34 | −0.42 | 0.03 | 70 | −0.35 | 0.02 |
| | | Gironde | 72 | 0.00 | 0.02 | 141 | −0.21 | 0.03 |
| | | Charente | 24 | 0.21 | 0.03 | 46 | 0.29 | 0.07 |
| | | Adour | 24 | −1.16 | 0.03 | 48 | −0.44 | 0.03 |
| | | Somme | 24 | 0.00 | 0.01 | 53 | 0.02 | 0.01 |
| | | Vilaine | 24 | 0.28 | 0.02 | 48 | 0.22 | 0.02 |
| NH$_4$ | mgN L$^{-1}$ | Seine | 150 | 0.04 | 0.13 | 299 | 0.16 | 0.07 |
| | | Loire | 34 | −0.20 | 0.02 | 70 | 0.13 | 0.02 |
| | | Gironde | 72 | 0.85 | 0.62 | 141 | 0.78 | 0.35 |
| | | Charente | 24 | 0.24 | 0.03 | 46 | 0.54 | 0.12 |
| | | Adour | 24 | 0.65 | 0.06 | 48 | 0.71 | 0.21 |
| | | Somme | 24 | −0.01 | 0.01 | 53 | 0.18 | 0.03 |
| | | Vilaine | 24 | 0.29 | 0.04 | 48 | 0.50 | 0.08 |
| NO$_3$ | mgN L$^{-1}$ | Seine | 150 | 0.06 | 0.70 | 298 | 0.05 | 0.55 |
| | | Loire | 34 | 0.20 | 1.08 | 70 | 0.15 | 0.67 |
| | | Gironde | 72 | 0.15 | 0.47 | 141 | 0.07 | 0.33 |
| | | Charente | 24 | 0.08 | 0.51 | 46 | 0.05 | 0.51 |
| | | Adour | 24 | −0.09 | 0.48 | 48 | 0.01 | 0.46 |
| | | Somme | 24 | −0.21 | 0.54 | 53 | −0.22 | 0.61 |
| | | Vilaine | 24 | −0.23 | 1.02 | 48 | −0.32 | 0.96 |
| DSi | mgSi L$^{-1}$ | Seine | 132 | 0.10 | 0.69 | 269 | 0.11 | 0.66 |
| | | Loire | 34 | −0.08 | 1.33 | 70 | −0.08 | 0.88 |
| | | Gironde | 36 | 0.22 | 1.60 | 103 | 0.15 | 1.10 |
| | | Charente | 0 | - | - | 2 | NR* | NR* |
| | | Adour | 14 | −0.43 | 0.95 | 26 | −0.16 | 0.74 |
| | | Somme | 24 | −0.19 | 0.83 | 51 | −0.20 | 0.88 |
| | | Vilaine | 24 | −0.19 | 0.47 | 48 | −0.15 | 0.63 |
| TOC | mgC L$^{-1}$ | Seine | 150 | 0.27 | 1.17 | 299 | 0.26 | 1.51 |
| | | Loire | 0 | - | - | 0 | - | - |
| | | Gironde | 42 | 0.27 | 1.67 | 90 | 0.25 | 1.20 |
| | | Charente | 24 | 0.35 | 2.77 | 46 | 0.18 | 1.08 |
| | | Adour | 12 | 0.39 | 1.11 | 24 | 0.39 | 1.22 |
| | | Somme | 11 | −0.06 | 0.23 | 24 | 0.01 | 0.22 |
| | | Vilaine | 0 | - | - | 0 | - | - |

* NR, not relevant in the case of a data set with fewer than 10 samples.

### 3.1.2. Validation for 2014 and 2016

After the calibration step performed on the year 2015, the model was implemented over the entire 2014–2016 period for validation. Figure 4 shows the performance of the C-GEM model over the entire period studied (from 2014 to 2016, including the calibration and validation periods; note that the validation results of the small-scale estuaries are presented in Figure S-2 in the SM). The seasonal variations are depicted (see Figure 4 and S-2) at specific monitoring stations located in the lower part of each estuary (but not too close to the marine boundary conditions in order to capture the dynamics of the model and minimize the influence of the marine boundary). Seasonal variations of other stations close to the marine boundary and within the salinity gradient are additionally presented in SM (Figure S-3) to offer more information on the model performance. A rather good evaluation of these water quality time series simulations is confirmed by the Bias and RMSE indicators, which take into account all the observations gathered (Table 7).

In agreement with the longitudinal profiles simulated for the calibration period (Figure 3), the seasonal $PO_4$ levels (simulated and observed) in the downstream Seine estuary (average, 0.11 mg P $L^{-1}$, e.g., 3.5 µmol $L^{-1}$) was twice as large as those for the other estuaries (ranging on average from 0.04~0.07 mg P $L^{-1}$, e.g., 1.3~2.3 µmol $L^{-1}$). Similar to $PO_4$, $NH_4$ also showed the highest levels in the Seine estuary, with large variations (0.01–0.35 mg N $L^{-1}$, e.g., 0.7-25.0 µmol $L^{-1}$). Similar large magnitudes for $NH_4$ concentrations were also found for the medium-size and small estuaries (Figure 4).

The $NO_3$ concentrations of the Seine (4.8 mg N $L^{-1}$, e.g., 342.9 µmol $L^{-1}$), the Charente (5.3 mg N $L^{-1}$, e.g., 378.6 µmol $L^{-1}$), and the Somme (4.7 mg N $L^{-1}$, e.g., 335.7 µmol $L^{-1}$) were ~2–4 times higher than the Loire (2.6 mg N $L^{-1}$, e.g., 185.7 µmol $L^{-1}$), Gironde (1.6 mg N $L^{-1}$, e.g., 114.3 µmol $L^{-1}$), Adour (1.4 mg N $L^{-1}$, e.g., 100.0 µmol $L^{-1}$), and Vilaine (2.3 mg N $L^{-1}$, e.g., 164.3 µmol $L^{-1}$); the DSi values of the Somme (9.0 mg Si $L^{-1}$, e.g., 321.4 µmol $L^{-1}$), Loire (5.1 mg Si $L^{-1}$, e.g., 182.1 µmol $L^{-1}$), and Charente (4.8 mg Si $L^{-1}$, e.g., 171.4 µmol $L^{-1}$) were ~1.4–3.3 times larger than in the other systems (2.7~3.5 mg Si $L^{-1}$, e.g., 96.4-125.0 µmol $L^{-1}$, Figure 4). The simulations of TOC in the Loire (3.7 mg C $L^{-1}$, e.g., 308.3 µmol $L^{-1}$) and Vilaine (2.8 mg C $L^{-1}$, e.g., 233.3 µmol $L^{-1}$) appeared larger than in the other systems, but no measured TOC values were available; TOC simulation levels in the Seine (2.3 mg C $L^{-1}$, e.g., 191.7 µmol $L^{-1}$), Charente (1.9 mg C $L^{-1}$, e.g., 158.3 µmol $L^{-1}$), Adour (1.5 mg C $L^{-1}$, e.g., 125.0 µmol $L^{-1}$), and Somme (2.6 mg C $L^{-1}$, e.g., 216.7 µmol $L^{-1}$) were in agreement with the observations. As shown by the highest algal biomass (Chl-a peaked to 40–50 µg $L^{-1}$), the Seine estuary was the most eutrophic system (Figure 4). All the estuaries studied seemed well-oxygenated during the study period (2014–2016). DO varied around the same range, with mean values from 8.0 to 9.9 mg $O_2$ $L^{-1}$ (around 70-100% saturation), but for the Charente, DO dropped to <5.0 mg $O_2$ $L^{-1}$ during July–September (<50% saturation), which was not well represented by the model (Figure 4).

In addition, the seasonal trends generated by the model for nutrients, DO and Chl-a concentrations generally follow the variations reported by field measurements both in timing and amplitude. $PO_4$ values were higher in summer. The $NO_3$ concentrations clearly showed a seasonal decrease from spring to autumn for most of the estuaries studied (Figure 4). DSi concentrations indicated lower values mostly in spring and late summer, linked to siliceous diatom uptake, which corresponded to Chl-a peaks, clearly visible in the Seine, Somme, and Vilaine. The simulation underestimated silica uptake in 2014 for the

Gironde. For the other estuaries, seasonal DSi patterns were not clear and data were often missing. The highest TOC level occurred in winter, with higher discharges and high SPM, although some high summer values can be linked to biological biomass. DO showed a regular trend with high values in winter and low values in summer, according to its solubility, but also to its consumption with high summer mineralization. However, whereas the Charente River showed DO observed values of about 11 mg $L^{-1}$ in winter (~100% saturation) much lower DO values down to 3-4 mg $O_2$ $L^{-1}$ were found in summer (i.e., ~35%

saturation, well illustrating a high summer $O_2$ consumption, but these low DO value were not well simulated by the model (Figure 4). Indeed, phytoplankton biomass (Chl-a) simulations showed a shift compared to the observations for 2014, which led to short-term summer DO peaks of the model and did not fit the observations. Noteworthy is the excessively scarce phytoplankton data which cannot support the modelled pattern at such a time scale.

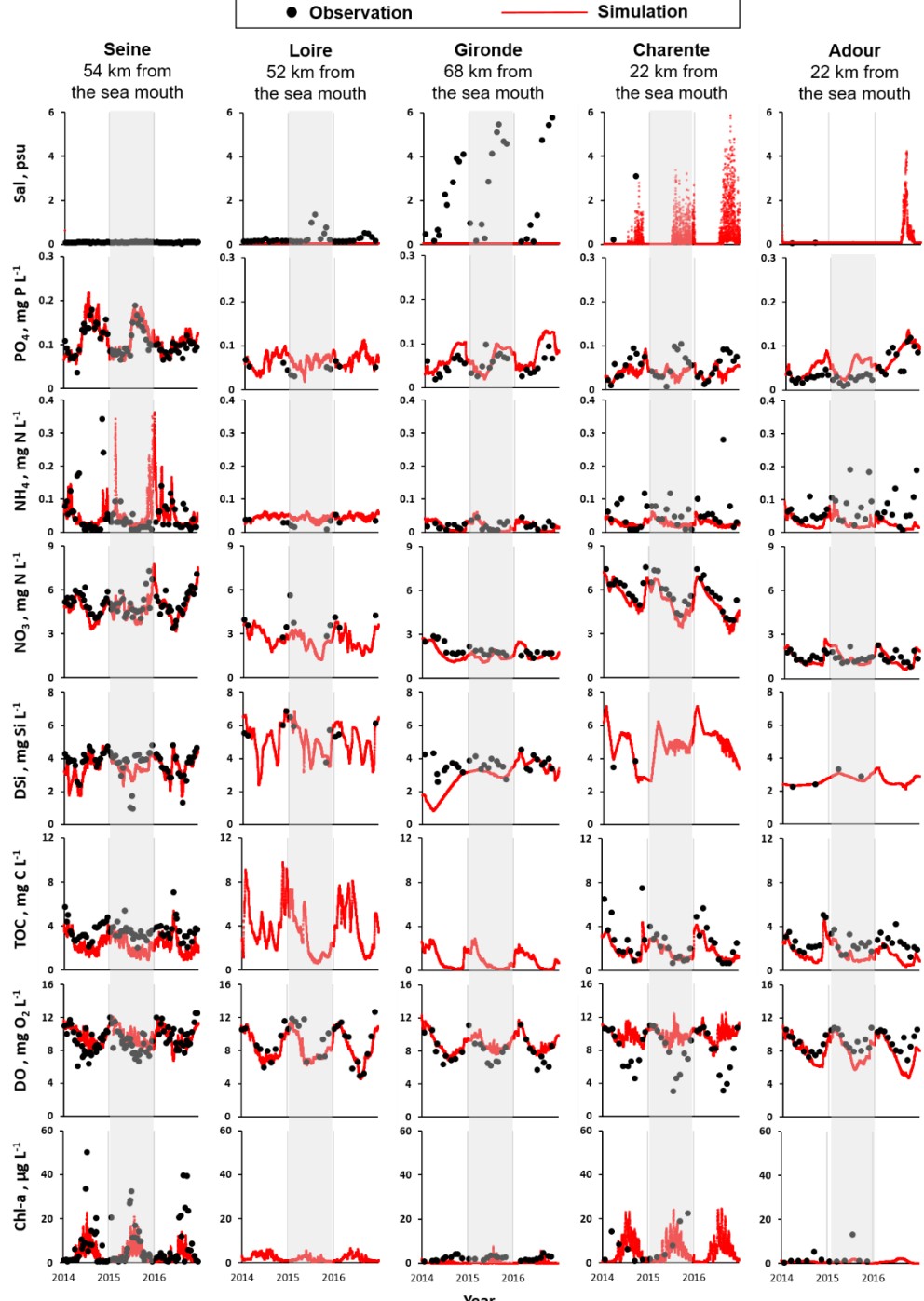

**Figure 4 Temporal variations for salinity (Sal), phosphate (PO₄), ammonium (NH₄), nitrate (NO₃), dissolved silica (DSi), total organic carbon (TOC), dissolved oxygen (DO), and chlorophyll a (Chl-a) concentrations from 2014 to 2016 for the Seine, Loire, Gironde, Charente, and Adour estuaries at the sampling stations located about 1/3 the length of the estuary to the sea mouth. Gray columns covered the year of calibration (2015). The results of the Somme and Vilaine are shown in the supplementary material.**

## 3.2. Biogeochemical Budgets

### 3.2.1. Import and Export Fluxes

Considering the suitable agreement between the levels of simulations and observations from the previous section, a reasonable level of confidence can be attributed to the import and export nutrient fluxes calculated by the model at the limits of the estuaries.

The import of TN ranged from 5.5 (Somme) to 104.6 kt N yr$^{-1}$ (Loire), TP from 0.1 (Somme) to 5.0 kt P yr$^{-1}$ (Loire), TSi from 9.4 (Somme) to 142.0 kt Si yr$^{-1}$ (Loire), Phy from 0.1 (Somme) to 3.1 kt C yr$^{-1}$ (Loire), and TOC from 2.8 (Somme) to 154.9 (Loire) kt C yr$^{-1}$. Although the river discharge of the Gironde estuary was slightly larger (2%) than that of the Loire, the Loire estuary received larger nutrient fluxes (1.7 times for TN, 2.6 times for TP, 15.3 times for TSi, 2.4 times for Phy, and 2.0 times for TOC) than the Gironde. Further, the Loire, with 1.6 times the discharge of the Seine estuary, imported only 1.2 times the TN flux to its estuary but 1.8 times the TP flux. The river discharge of the Adour was quite similar to the Seine (86% of the Seine), but its TN, TP, TSi, Phy, and TOC imports amounted to only 25%, 34%, 47%, 24%, and 40%, respectively, of those of the Seine (Table 8).

Based on this study, around 259 kt N yr$^{-1}$ of TN, 9.6 kt P yr$^{-1}$ of TP, 304 kt Si yr$^{-1}$ of TSi, and 145 kt C yr$^{-1}$ of TOC were exported to the French Atlantic Ocean through these seven estuaries. They accounted for about 80% of the total water discharge from all the estuaries on the French Atlantic coast (based on long-term analysis of runoff data from French national databases) and 83% of the total watershed areas on the French Atlantic coast. However, considering that the retention rates seem to decrease with the size of the systems, it is likely that the riverine loads carried by the small estuaries not taken into account in our study are only marginally affected by the estuarine filter function. As a consequence, the cumulated nutrient export to the sea of the 7 estuaries investigated in our study might be slightly lower than the 80% representing the total water fluxes.

**Table 8 The mean annual (2014–2016) import and export of nutrients (total nitrate (TN), total phosphorus (TP), total silicate (TSi), phytoplankton (Phy), and total organic carbon (TOC)) for the estuaries studied. The export fluxes are calculated by the model, while import ones are observed after interpolation of the water quality variables.**

| Variables | | Unit | Large-scale | | | Medium-scale | | Small-scale | |
|---|---|---|---|---|---|---|---|---|---|
| | | | Seine | Loire | Gironde | Charente | Adour | Somme | Vilaine |
| Annual mean Q | | m$^3$ s$^{-1}$ | 499 | 808 | 824 | 77 | 431 | 48 | 96 |
| Riverine basin area | | km$^2$ | 65,000 | 111,436 | 75,125 | 7598 | 14,832 | 5560 | 10,498 |
| Estuarine basin area | | km$^2$ | 11,843 | 6891 | 7677 | 2327 | 2122 | 820 | 237 |
| Whole basin area | | km$^2$ | 76,843 | 118,327 | 82,802 | 9925 | 16,954 | 6380 | 10,735 |
| Estuarine fresh water residence time | | day | 16 | 20 | 79 | 12 | 2 | 17 | 8 |
| Export | TN | kt N yr$^{-1}$ | 78.5 | 69.0 | 46.5 | 13.9 | 19.1 | 5.4 | 26.7 |
| | TP | kt P yr$^{-1}$ | 2.3 | 3.9 | 1.4 | 0.2 | 0.9 | 0.1 | 0.9 |
| | TSi | kt Si yr$^{-1}$ | 52.3 | 133.2 | 56.2 | 11.5 | 26.0 | 8.9 | 16.0 |
| | Phy | kt C yr$^{-1}$ | 11.6 | 3.6 | 1.9 | 0.1 | 0.2 | 0.4 | 1.2 |
| | TOC | kt C yr$^{-1}$ | 25.1 | 64.0 | 3.4 | 4.8 | 16.0 | 2.5 | 29.8 |
| Import | TN | kt N yr$^{-1}$ | 84.9 | 104.6 | 61.4 | 14.8 | 21.3 | 5.5 | 26.2 |
| | TP | kt P yr$^{-1}$ | 2.8 | 5.0 | 2.0 | 0.2 | 0.9 | 0.1 | 0.9 |
| | TSi | kt Si yr$^{-1}$ | 56.0 | 141.9 | 90.3 | 12.0 | 26.4 | 9.4 | 16.1 |
| | Phy | kt C yr$^{-1}$ | 1.0 | 3.1 | 1.3 | 0.1 | 0.2 | 0.1 | 1.2 |
| | TOC | kt C yr$^{-1}$ | 56.7 | 154.9 | 75.9 | 6.8 | 22.5 | 2.8 | 31.5 |

Note:

DIN=NH$_4$+NO$_3$, TN=DIN+(Phy (in C)+TOC)/5.7 (the Redfield–Brzezinkski ratio, Brzezinski, 1985; Redfield et al., 1963)
DIP=PO$_4$, TP=DIP+(Phy (in C)+TOC)/41 (the Redfield–Brzezinkski ratio, Brzezinski, 1985; Redfield et al., 1963)
TSi=DSi+Phy (in C)/2.8 (the Redfield–Brzezinkski ratio, Brzezinski, 1985; Redfield et al., 1963)

Interestingly, considering the specific TP, TN, and TOC fluxes to the estuaries (i.e., total influxes per surface area of river basin) makes it possible to infer the contamination in the watersheds. The Vilaine and the Charente estuaries receive the highest fluxes of TN, while the Vilaine and Adour estuaries were the receptacles for greater TP and TOC fluxes with respect to the
size of their upstream watershed (Figure 5). Although silica is a major nutrient for diatom algae populations it is of natural origin (rock weathering), differently from anthropogenic N and P sources, and reveals the lithology of the watershed, with higher fluxes for the Adour and Somme estuaries (Figure 5). Overall TSi specific fluxes exceeded 850 kg Si km$^{-2}$ yr$^{-1}$, varying within a range of 2.1 times, while TP and TOC specific fluxes were more variable, within a factor close to 5~6 and TN specific fluxes within a range of 2.5 times (Figure 5).

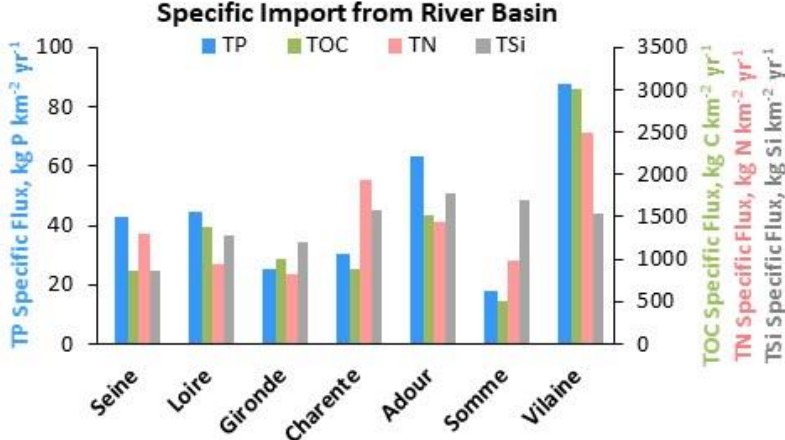

**Figure 5 Average annual (2014–2016) specific fluxes of total phosphorus (TP), nitrogen (TN), organic carbon (TOC), and silica (TSi) entering the estuaries studied.**

### 3.2.2.    Retention Rate

To determine the different ecological functions of the estuaries studied, their retention rates for TN, TP, TSi, and TOC were
quantified following the equations presented in section 2.4 (Figure 6).

For the large estuaries (the Seine, Loire, and Gironde), the retention rate for TN ranged from 8% to 34%, for TP from 19% to 27%, for TSi from 6% to 39%, for TOC from 56% to 96%. For the medium-size estuaries (the Charente and Adour), the retention rate was 7–10% for TN, 1–12% for TP, 2–7% for TSi, and 30–32% for TOC. For the small estuaries (length <20 km, the Somme and Vilaine), the retention rate for TN, TP, TSi, and TOC were 0–1%, 0–7%, 2–6%, and 8–12%, respectively.
The Loire estuary showed the largest retention rate for TN; the Gironde estuary is the most retentive for TP, TSi, and TOC.

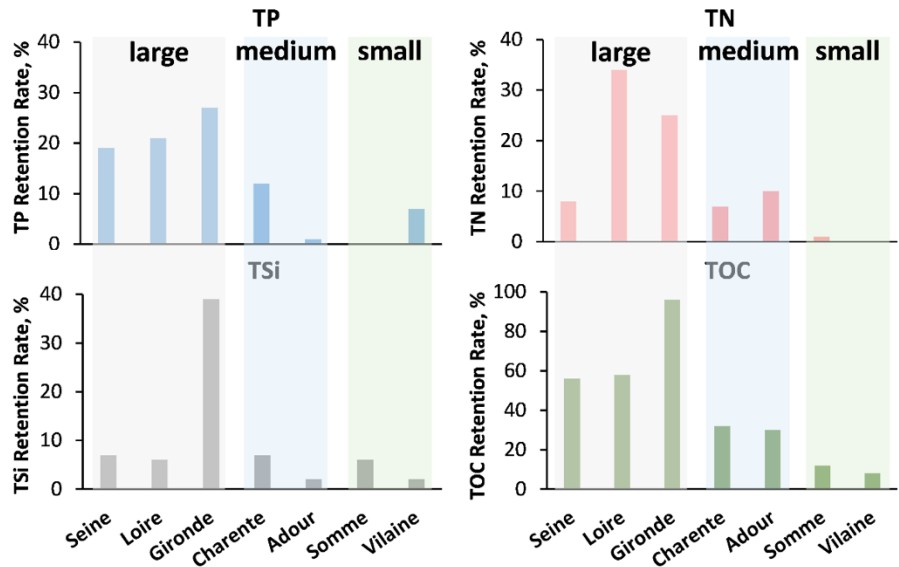

**Figure 6 Annual mean (2014–2016) retention rates for the seven estuaries studied in terms of total phosphorus (TP), nitrogen (TN), organic carbon (TOC), and silica (TSi).**

For the three large estuaries, the annual mean water residence time was around 14–27 days for the Seine and Loire, but longer
450 for the Gironde estuary: 70–97 days. Longer water residence time indeed causes higher retention rates for the Gironde (Figure 6 and Figure 7). For the two medium-size estuaries, the water residence time differed substantially: for the Charente, it ranged from 9 to 18 days while the water residence time of the Adour was only 2–3 days during the study period. The Adour estuary had a shorter estuarine length and was narrower but with an approximately six times greater river discharge compared to the Charente. Thus, the retention rate for the Charente estuary was also higher (Figure 6 and Figure 7). Small estuaries such as the
455 Somme and Vilaine had a water residence time ranging from 5 to 18 days. The Vilaine had a short estuarine length due to the dam located 10 km from the sea mouth. Therefore, the water residence time of the Vilaine estuary depends on the dam's regulation and the residence time of the dam/reservoir at the interface of the river outlet and the estuary (Figure 6 and Figure 7).

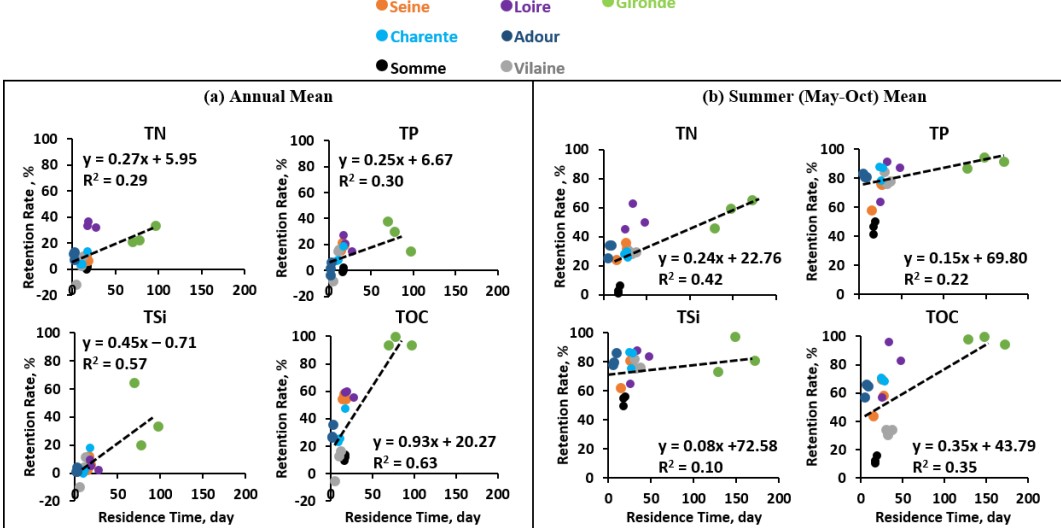

**Figure 7 The relations between the annual retention rate for nutrients (total nitrogen (TN), total phosphorus (TP), total silica (TSi), and total organic carbon (TOC)) and the fresh water residence time for the estuaries studied (the Somme, Seine, Vilaine, Loire, Charente, Gironde, and Adour estuaries) for the 3 years 2014–2016. (a). Annual mean; (b). Summer (May-October) mean.**

We therefore plotted the relationship between annual retention rates for water quality variables (TN, TP, TSi, and TOC) and annual mean water residence time for 2014–2016 for each estuary studied (Figure 7). The retention rates for TSi and TOC showed a significant positive relationship ($R2=\sim0.6$, $p<0.05$, Figure 7) with annual mean water residence time, while the positive relationship was less clear for TN and TP retention rates. We also calculated retention rates and residence times for summer season (May-Oct) which further showed that the estuaries had larger residence time during the dry season, which also led to larger retention rates. The relationships are however not strictly linear, % retention plateauing at high residence times. The reaction process rates calculated by the model along the estuaries over the 3 years were integrated and provided as annual average, in ton of total organic Carbon, N-NO3, dissolved Si or phosphate per year for organic matter degradation, denitrification, net primary production either from diatoms or the from all the algal community, respectively (see Table S-1 in supplementary material). These fluxes are highest in the largest estuaries with the largest carbon and nutrients loads as well as the highest residence time. Reported in terms of percentage of the import fluxes, these values allow comparing the intensity of the biogeochemical processing simulated by the model between the different estuaries as well as with carbon and nutrients retention rates discussed above. Overall, Total organic carbon degradation and NO3 denitrification percentages were the highest in the Gironde (98 % and 26%, respectively) and the lowest for the Vilaine (10%, 3%, respectively). Other systems displayed intermediate values falling in the 24-39% and 3-14% ranges for TOC degradation and denitrification, respectively (Table S-1). This intensity of organic carbon processing is consistent with the global figures suggested at the global scale by Bauer et al. (2013) as well as previous modeling studies performed with C-GEM in Europe and along the East coast of the US (Volta et al., 2016, Laruelle et al., 2017, 2019). The integrated denitrification rates are also consistent with the compilation performed by Nixon et al (1996). Biogeochemical reactivity regarding organic matter degradation and denitrification appeared greater for the most retentive Gironde, with its longest residence time. This trend was also illustrated by the calculation of mixing curves for the Seine, Loire and the Gironde (the 3 systems with the longest salinity gradient) computed for the reference simulation and simulations in which the biogeochemical model of C-GEM was deactivated (Figure S2 of the supplementary material). The difference between the reference simulation and the simulation without biogeochemistry provides a visual

representation of the intensity of the biogeochemical processing within the system and is significantly more pronounced in the Gironde. Interestingly, silica uptake percentages, which are entirely sustained by diatoms primary production were also the highest in the Gironde, the Seine and the Somme (16%, 13% and 12%, respectively), in accordance with the ones for phosphates (16%, 43%, 76%, respectively). These results revealed however a large range of autotrophic activity. The percentage PO4 uptake flux was particularly high in the Somme because of its proportionally low specific P riverine loads from its upstream boundary. Overall biogeochemical phosphates fluxes were rather well balanced with those of silica, according to the Redfield ratios, indicating mostly a development of diatoms (Table S-1).

### 3.3. The Indicator of Coastal Eutrophication Potential

The ICEP was calculated on the basis of estuarine TN, TP, and TSi deliveries (Figure 8). The N-ICEP values at the sea mouth were all positive and ranged from 2.1 (Loire) to 14.5 kg C km$^{-2}$ d$^{-1}$ (Charente), while the P-ICEPs at the sea mouth were all negative, ranging from −8.7 (Adour) to −3.8 kg C km$^{-2}$ d$^{-1}$ (Seine). The N-ICEPs were positive and the P-ICEPs were negative for all the estuaries studied, indicating that TN was in excess relative to TSi, revealing a risk of coastal eutrophication and the development of potentially harmful non-siliceous algae. Regarding the P-ICEP, the systematic negative values would rule out any risk of eutrophication, although the P-ICEP closest to zero for the Seine and the Gironde would indicate that these systems are fragile, given that increased P in case of low discharge could become positive and lead to eutrophication.

Referring to the potential of coastal marine eutrophication, we calculated the P- and N-ICEP at the entrance and outlet of the estuary, to determine the buffer role of estuaries regarding eutrophication potential (Figure 8). Whereas the P-ICEP was less negative for export fluxes compared to the import ones, the N-ICEP was generally lower for the export fluxes, meaning that at the coastal zone, there was a lower deficit in phosphorus compared to silica, and a reduced excess in nitrogen, except for the Vilaine for which the P- and N-ICEPs were similar for both the export and import fluxes.

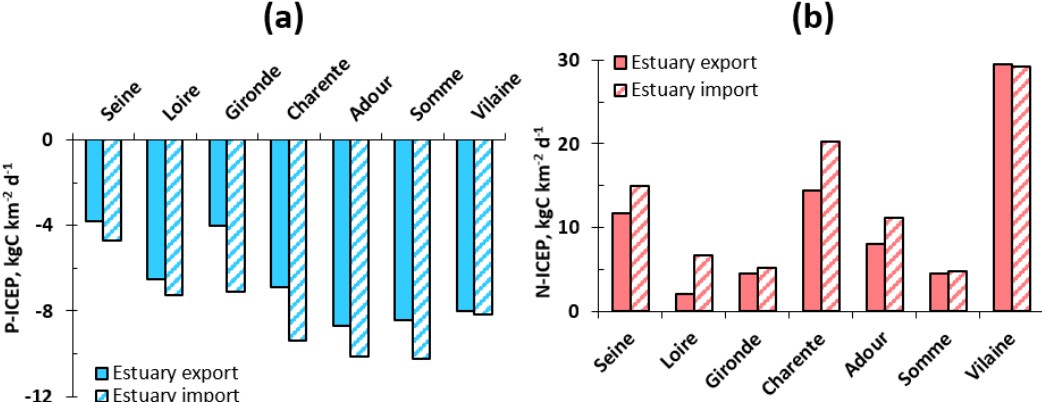

**Figure 8 Indicator of costal eutrophication potential (ICEP, (a): P-ICEP; (b): N-ICEP) for the estuaries studied.**

## 4. Discussion

### 4.1. Model Applicability and Limitations

The 1-D biogeochemical model C-GEM was built to overcome the requirement of large, often unavailable data sets (e.g., geometry at a fine resolution) needed for implementing complex multidimensional models and to improve computation efficiency and consequently enable regional- and/or global-scale applications using a generic, theoretical framework based on the direct relationship between estuarine geometry and hydrodynamics (Volta et al., 2014, 2016b). As mentioned above, this requires easily obtainable geometrical parameters to frame the idealized morphology of the estuary and uses a set of

exponential functions to estimate the width profile of the channel and the cross-section variations along the longitudinal axis. The sensitivity of biogeochemical processes to uncertainties in parameter values provided by Volta et al. (2014a) revealed on large estuaries that the convergence length and water depth had an important influence on the biogeochemical functioning of the estuary due to their strong constraint over the hydrodynamics and transport of solids within the system (Savenije, 2012). We could thus expect a less satisfactory performance on small-scale estuaries (Somme and Vilaine, Table 7, Figure S-1 and

Figure S-2 in the SM) than on medium- and large-scale estuaries. Indeed, the spatial resolution used in C-GEM is 2 km, which allows capturing the dominant features of large estuaries while it can be too coarse for small estuaries (length <30 km) of this study. Although we aimed to implement a similar version of C-GEM for all the studied estuaries here, for future specific applications on small estuaries, the grid size might need to be adjusted.

In its current setup, the biogeochemical module of C-GEM considers some of the most essential biogeochemical processes

and reactions (i.e. primary production, organic matter degradation, denitrification…). In spite of generally good ability of the model to capture the main spatial and temporal biogeochemical dynamics of the different systems studied (i.e., longitudinal, seasonal and amplitude of the variations of nutrients carbon and oxygen fields), several potentially important processes contributing to the N and P cycling in estuarine environments in particular are still ignored or largely simplified. These include benthic–pelagic exchanges, sorption–desorption of phosphorus, mineral precipitation or a more complex representation of the

biological planktonic/benthic compartments (such as grazing by higher trophic levels, or multiple reactive organic carbon pools for instance). With its current set-up, the lack of explicit benthic biogeochemical module obviously limits the depth of mechanistic understanding the model can provide of nutrient cycling, particularly regarding interactions between pelagic and benthic compartments which can significantly influence the intensity but also the timing of nutrients and organic matter cycling in estuaries (Laruelle et al., 2019). In that context, future developments of the model should include the implementation of

several benthic processes this task comes with a number of hurdles. For instance, while the addition of a full diagenetic module at each grid cell of our model would be possible, it would also increase its calculation time by one order of magnitude and likely require a very long spin-up to generate initial conditions for the benthic species. There exist simpler benthic modules of lower complexity, which would limit the computation cost of adding an explicit representation of benthic processes (Billen et al., 2015; Soetaert et al., 2000) but those would nonetheless significantly increase the data demand of the model to be properly

calibrated. Indeed, the increase in complexity of the model will involve the use of field data to constrain and calibrate the

newly implemented processed. While measurements of estuarine benthic processing of nutrient and carbon do exist, they are still relatively scarce. In the present study, the simple representation of particulate matter burial that was implemented and applied to phytoplankton and TOC to provide a first-order representation of the process, which is necessary to evaluate the retention of carbon and nutrients within the system. We believe this addition, coupled with denitrification provides a first insight on the main pathways removing nutrients from estuaries and allows calculating carbon and nutrient retention rates that can be compared with previously published estimates

In addition to the idealized geometry and biogeochemical processes, the boundary conditions and constraints are also critical for the performance of the model because they place the simulated system in an environmental context and drive transient dynamics. In this study, marine boundary conditions were extracted from ECO-MARS3D (Cugier et al., 2005b; Lazure and Dumas, 2008; Ménesguen et al., 2019), and are thus model-derived, but its robustness has been proved for both hydrodynamic and biogeochemical variables in the marine coastal zones. Riverine boundary conditions comprise the observed data, extracted at the station closest to the model's upper boundary, which might still be located several kilometers upstream or downstream of the model's upstream boundary. Noteworthy, the Dordogne River which was considered as a source of water and biogeochemical material for the Gironde estuary at the confluence, that ignored the tidal cycle effects that rise in the Dordogne River.

Observed data are usually measured at monthly or bimonthly intervals. Some variables were however sampled more frequently, two or three times per month, while Chl-a concentrations were usually only available from March to September. Also, DSi and TOC are usually less available than other variables. Instantaneous SPM and water quality sampling data were linearly interpolated to obtain the daily values between the adjacent measurements. Therefore, simulations are only partly transient and do not resolve events such as storms, floods, and/or extreme droughts.

In this study, only WWTPs from the largest estuarine cities (>50,000 inhabitants) were considered in the estuarine model. The volume of water released by WWTPs to the estuaries may change over a year, especially in the touristic estuarine cities in summer. However, even if the summer fluxes double and the summer river discharges decrease by half compared to the average values, wastewater discharge would remain below 3% of the total.

Although these simplifications and limitations, e.g., morphological and boundary inputs, C-GEM captured the right level of the variables and the main spatial and seasonal trends, also considering potential inaccuracies from the sampling strategy (surface sampling, ebb or flow, etc.). Its performance is supported by simulation/observation comparisons with longitudinal profiles from specific days (Figure 3 and Figure S-1 in the SM), and/or simulation/observation comparisons at specific cross sections (Figure 4 and Figure S-2), and/or evaluation analysis (Table 7). Further, solid results were gained elsewhere with C-GEM supporting its genericity, i.e. carbon processing in the six major tidal estuaries (length >80 km) flowing into the North Sea (Volta et al., 2016b), biogeochemical dynamics and $CO_2$ exchange in three tidal estuaries (length >90 km, Volta et al. 2016a), $CO_2$ evasion on 42 tidal estuaries along the US east coast (Laruelle et al., 2017) and the Seine (Laruelle et al., 2019), biogeochemical processes and fluxes on a tropical estuary (Nguyen et al., 2021).

## 4.2. Fluxes and Retention Capacity

Quantifying fluxes (section 3.2.1) showed that even though some estuaries received nearly the same water flow from the upstream basin, their nutrient imports can differ greatly, due to the range of the human population and land use in the upstream watersheds. This is illustrated, for example by the Seine and Loire compared with the Adour. Anthropogenic influences have been recognized as an important factor affecting water quality as well as nutrient fluxes not only in rivers from headwater to downstream estuarine and coastal waters, but also in stagnant systems (Baker, 2005; Escolano et al., 2018; Garnier et al., 2019;

Nguyen et al., 2021). In this study, the basins are characterized by different land use types. For example, the Somme, Seine, Vilaine, and Charente basins, with the highest nitrate concentrations are dominated by intensive agriculture with arable land covering 66–84% of the total basin area (Table 2). Intensive agriculture is indeed well known for nitrate contamination of ground- and surface waters (Billy et al., 2013; Lockhart et al., 2013). Moreover, the upstream basin of the Seine, Loire and Gironde estuaries have a population equaling 16.7, 8.3 and 4.9 $10^6$ inhabitants, respectively, the one of the Seine basin being

by far the highest. Despite treatments of most of the domestic effluents have been improved since the Urban Wastewater Treatment Directive (1991) and the EU Water Framework Directive (WFD, 2000), high levels of $NH_4$ and $PO_4$ still remains good indicators for domestic pollution. In the Seine, these levels have been considerably reduced (Garnier et al., 2019), but still remains rather high compared to the other rivers.

The positive relationship between retention rates and water residence time confirms that estuarine geomorphology and river

fluxes are major physical forcings for biogeochemical retention abilities (Arndt et al., 2009; Perez et al., 2011; Romero et al., 2019). This relationship between estuarine residence time and retention capacity was already theorized empirically on a handful of systems in the pioneering work of Nixon et al. (1996). While deriving a predictive formula for nutrient retention solely based on geometric parameters and nutrient loads remains elusive (Laruelle et al., 2017), this study presents further evidence of the importance of water residence time in the complex interplay of physical and biogeochemical drivers constraining the

retention potential of a given estuarine system. The role of residence time on nutrient retention/elimination is not only the fate of estuaries but also of rivers, stagnant systems (lakes, ponds and reservoirs), and also wetlands. This function can be valued for reducing contaminations and sometimes even promoted for restauration, although the best way is to limit the amount of fertilizer applied (Bernot and Dodds, 2005). The retention rates for TSi and TOC showed relatively significant positive relationships (Figure 7) with annual mean water residence time, while relationships between TN and TP retention rates and

the annual mean water residence time were weaker. This can be caused by anthropogenic interferences with the biogeochemical processes within the estuaries. For example, the high retention for TN in the Loire estuary might be due to an elimination process in this system (e.g. denitrification, see Table S-2 in SM). For the Vilaine basin, 2014 was a wet year with a discharge double those recorded in 2015 and 2016, leading to slightly negative retention rates, meaning that instead of eliminating nutrients, the Vilaine estuary exported a small amount (Figure 7). The export of nutrients likely corresponds to a difference in

the nutrient stock within the system itself between the beginning and the end of the simulated period and remains limited to a few percent of the riverine loads. Overall, TN and TP retention rates are comparable in the estuaries simulated, but some

systems are more efficient at removing P while other are more efficient at removing N. This implies that the TN/TP ratio varies along the estuarine gradient. These variations are controlled by the complex interplay of phytoplankton and organic matter (TOC) burial and denitrification along the estuary. Denitrification, being a net removal of N obviously increases the TN:TP ratio. However, the effect of phytoplankton and organic matter burial is more subtle. In the model, it is assumed that the material buried (phytoplankton and organic matter) has a fixed Redfield N:P ratio and thus removes the same proportion of N and P through burial. However, the ratio of inorganic (DIN or DIP) to organic matter (associated with phytoplankton or TOC) is different for N and P. As a consequence, the net effect of burial also affects the TN:TP ratio of the system.

The retention rates were also compared with other estuaries. A yearly retention rate of 6.8–13% for TN, 27–31% for TP, and 4.3–11% for DSi were found for the Seine estuary (Garnier et al., 2010b; Romero et al., 2019), while we found here, for the 2014–2016 period, values of 6–10% for TN, 15–21% for TP, and 2–12% for TSi, which agree closely with the earlier studies. For tidal estuaries discharging into the North Sea, a 15% retention rate for total C was found (Volta et al., 2016b), and the Scheldt estuary showed a 73% and 78% reduction of $NH_4$ and TOC, respectively (Vanderborght et al., 2007), and a retention rate of 12% for DSi and 32% for TN. Seitzinger (1988) found a 20–50% elimination of N for six estuaries, while for a tropical system, Luu et al. (2012) mentioned a 43.6% retention rate for N. Interestingly, the retention rates for individual systems are highly variable (Arndt et al., 2009). Here, TOC retention rates varied greatly among different systems, from 10% (the Somme estuary in 2014) to 100% (the Gironde estuary in 2016). With the longest water residence time and the high level of suspended matter and associated TOC, the Gironde is a site for high TOC retention, as already shown by Etcheber et al. (2007). TN filter capacities ranged from 0% (the Somme estuary in 2014) to 37% (the Gironde estuary in 2016), and in the Gironde was linked to denitrification (26%, see Table S-2 in SM). Regarding the lack of retention in some estuaries might mean that $NO_3$ production (e.g., by nitrification) would be compensated by losses (e.g., denitrification) as already observed in the Seine (Sebilo et al., 2006). TSi was eliminated only slightly (e.g.,, 2% for the Seine estuary in 2016 and the Loire in 2015 and 1% for the Adour in 2014 and 2015), suggesting that no diatom uptake occurred or was compensated by freshwater diatom mortality sensitive to the salinity gradients (Ragueneau et al., 2002; Roubeix et al., 2008). High DSi retention in the Gironde (39%) may be rather linked to detrital particle removal by burial, because turbid estuaries showed low diatom planktonic primary production with high suspended matter and low light penetration (Coynel et al., 2005).

Similarly to retention rates, intensities of the biological processes were clearly higher for larger estuaries with high residence times, especially for TOC and $NO_3$ removal, via total organic matter degradation and denitrification, largest estuaries behaving as receptacle of the upstream inputs and reactors for biogeochemical transformations (Howarth et al., 2011). DSi and phosphates removal intensities via net primary production were more variable between large and small estuaries, as well as the retention rates. High biogeochemical reactivity of estuaries in terms of primary production does not necessarily leads to high retention, if accompanied by remineralization.

### 4.3. Coastal Eutrophication Potential

Eutrophication has been recognized as a serious environmental threat for both rivers and marine waters. The amount of nutrients (N, P, Si) delivered to the coastal zone by large river systems are the major determinants of coastal marine eutrophication problems. Previous studies have indicated that eutrophication is not only a result of high inputs of anthropogenic nutrients (N and/or P), but also of the imbalance when anthropogenic N and P are introduced over Si from natural rock weathering (Billen and Garnier, 1997; Garnier et al., 2021). Therefore, the ICEP considers the balance of these nutrients (N,

P, Si) to calculate the coastal eutrophication potential in watershed area units (kg C km$^{-2}$ day$^{-1}$) for comparisons between the P- and N-ICEP (C unit) and among systems (per km$^2$).

The P-ICEPs were all negative while the N-ICEPs were all positive for the estuaries studied herein, indicating a deficit in P and excess N, respectively, with respect to the Si requirements of diatoms (Billen and Garnier, 2007). The negative values for the P-ICEP for small to medium-size estuaries are larger than for large-scale estuaries (Figure 8). Accordingly, there would be

no risk for potential coastal eutrophication regarding P; however, the rapid cycling of this element in coastal marine systems generally prevents it from becoming limiting (Kobori and Taga, 1979; Labry et al., 2016). Conversely, positive N-ICEP values might be responsible for the eutrophication problems observed and did not relate to estuary size. While exploring the N-ICEP over the long term (from 1950 to the 2010s), Garnier et al. (2010a) found an increase as a function of population density and rapid urbanization leading to the development of sewerage systems without sufficient treatment of sewage water, and also with

rapidly increasing agricultural production.

The manifestations of eutrophication along the coast are diverse. *Phaeocystis* blooms have been reported in the Somme Bight (Lamy et al., 2006; Lefebvre et al., 2011), while *Dinophysis* (dinoflagellates) and *Pseudo-nitzschia* (diatoms) are the dominant harmful algae in the Seine Bight (Cugier et al., 2005a; Garnier et al., 2019; Ménesguen et al., 2019). Whereas most of the rivers have shown a decrease in phosphorus in the last two decades (Romero et al., 2013), leading to a negative P-ICEP, as

found here, algal biomass increased (diatoms and dinoflagellates) with a shift in the peak in the Vilaine Bight (Ratmaya et al., 2019), mostly due to N fluxes brought by the Loire River, which flows northwards with the currents (Ménesguen et al., 2018a, 2019) and to internal sources from sediments (Ratmaya et al., 2019). No strong eutrophication problems have been reported off the Loire estuary, in the North Biscay Bay, without major changes during the last two decades, in contrast to the North of France (Gohin et al., 2019). Indeed high winter nutrient fluxes are accompanied by high suspended solids, preventing algal

growth (Guillaud et al., 2008).

Further, a positive N-ICEP in the Southern Biscay Bay coast did not lead to eutrophication problems at the coast, because the Gironde estuarine water fluxes are driven to the middle of the bay and those of the Adour spread throughout the Basque Country, especially at higher discharges (Ménesguen et al., 2018a). However, for the Charente estuary, which flows into the close Marennes-Oléron Bight, well known for oyster production, chronic summer mortality of "juveniles" was reported in

the late 1990s to the early 2000s (Soletchnik et al., 2007). Garnier et al. (2021) demonstrated that the manifestations of eutrophication at the coast of riverine deliveries (and the associated ICEP) not only depend on the nutrient fluxes and their

stoichiometry, but also on the morphology of the receiving media.

## 5. Conclusions

The 1-D biogeochemical model (C-GEM) was applied to seven estuaries with different sizes and morphologies along the
French Atlantic coast (the Somme, Seine, Vilaine, Loire, Charente, Gironde, and Adour). Transient simulations were
implemented on water quality variables ($PO_4$, $NH_4$, $NO_3$, DSi, TOC, DO, and Chl-a) for 3 years (2014–2016). The model was
calibrated (2015) and validated (2014, 2016) by comparing it with in situ sampling data along the estuaries. The results showed
that this model presented accurate descriptions of the hydrodynamics, transport, and biogeochemistry in these tidal estuaries,
using simplified representations of the estuarine geometry and the same set of biogeochemical parameters for all estuaries.
C-GEM also quantifies the nutrient fluxes imported to and exported by the estuaries, reflecting human activities in the upstream
watersheds (population, agriculture). The retention rates for TP, TN, DSi, and TOC for the estuaries studied showed that large
estuaries generally had a higher retention rate due to their longer water residence times. Longer residence time provides enough
time for biogeochemical reactions, thus reducing the fluxes of nutrients delivered to the sea. However, eutrophication does not
only depend on nutrient fluxes, but also on their balance or imbalance. Therefore, the indicator for coastal eutrophication
(ICEP) was evaluated for these seven estuaries, and indicated that for these estuaries, although no risk appears regarding P,
there might be a risk of coastal eutrophication due to the excess TN over DSi, and thus the development of potentially harmful
non-siliceous algae.

The present study thus provides a new understanding of the complex biogeochemical behavior of a range of estuaries. C-GEM
can be combined with river models to simulate future scenarios with different degrees of anthropogenic impacts on the
upstream basins, and, in turn, to assess how the estuaries would respond to potential or forthcoming disturbances. However,
ongoing work for a better representation of some biogeochemical processes (especially the water-sediment interface) should
be implemented, while a systematic study regarding the grid size could be carried out on large and small estuaries.

*Data availability.* The measured data used in this study can be accessed through the links in section 2.2. C-GEM source code
is available upon reasonable request to the corresponding authors.

*Supplement.*

*Author contribution.* XW constructed the data set, performed the simulations, analyzed the data, prepared the figures and
wrote the paper. JG GGL conceptualized and supervised this study, provided data, and completed, modified and revised the
paper. VT, PP and RLG provided data for the dataset used for this study and revised the paper. GB and MA revised this paper.

*Competing interests.* The authors declare that they have no conflict of interest.

*Acknowledgments.* The authors thank Jack Middelburg for editing the manuscript. Our grateful acknowledgements also go
to Pierre Anschutz and two anonymous reviewers for their constructive comments and suggestions.

*Financial support.* This study (Prest'Eaux project) was supported by the Office Français de la Biodiversité (OFB), greatly

acknowledged for the project follow-up. Xi Wei's post-doc was funded by this project. Josette Garnier, Research Director at the CNRS, was the PI of Prest'Eaux. Goulven G. Laruelle is a research associate of the FRS-FNRS at the Université Libre de Bruxelles and contributed to the development of G-GEM.

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
