# Peer review of "Nutrient transport and transformation in macrotidal estuaries of the French Atlantic coast: a modelling approach using C-GEM"

_Biogeosciences, 2021_

## Author Response (AR1)

We greatly thank all the reviewers for their thorough and helpful comments which contribute to improving the manuscript.

Please find our point-by-point responses (in blue) in the following, and changes in the manuscript are *in italic* here.

**RC1: Pierre ANSCHUTZ**

It is always very interesting to test the knowledge we have about natural systems by translating them into equations in mathematical models. This allows first to test the validity of the equated processes and second to build scenarios according to environmental changes. This manuscript presents results of nutrient retention in estuaries of the French Atlantic coast obtained with a mathematical model entitled C-GEM.

Macrotidal estuaries are complex transitional environments. Modeling nutrients in these systems is a challenge, because the biogeochemical processes are numerous and complex and the physics of estuaries alone is the subject of complex 3D models. Here the choice was made to build a simplified one-dimensional physical model coupled with biogeochemical reactions involving dissolved N, P and Si compounds, as well as suspended particles. This model is applied to 7 estuaries of different sizes and the model outputs are compared to the available data for the tested period. The model could be criticized for being too simple (1-D), but the biogeochemical part is relatively complete and in the end the results are promising. This 1-D approach is a sufficiently precise step to obtain interesting results. I have a few remarks that concern the validation of the model, the tidal cycles, and points of detail described below.

We thank Prof. Pierre Anschutz for reviewing this paper and providing constructive and detailed comments. We revised the manuscript according to the specific comments below.

**RC1.1:** Line 40: It is conventional to write that estuaries, and more generally water bodies, are facing increasing anthropogenic impacts. For European estuaries, it would have been fair to write this in the late 1990s. Since then, there have been major efforts to improve water quality. Nitrate and phosphate levels are much lower today than 30 years ago. It would be more accurate to write something like "despite efforts to improve natural water quality since the 2000s, estuaries remain receptacles for nutrients and contaminants..."

**AC1.1:** Thank you for pointing this out. This remark is correct and we made the necessary change on Line 40 following your suggestion:

*"Estuarine and coastal ecosystems throughout the world are some of the most heavily used and threatened natural systems (Barbier et al., 2011) despite the efforts to improve natural water quality since the implementation of the WFD in the early 2000s (Water Framework Directive, EU-WFD 2000). While some improvements have been observed in recent decades regarding some specific perturbations such as, for instance, a general decrease in riverine phosphorus loads across Europe (Romero et al., 2013; Grizzetti et al., 2012), estuaries are still facing significant anthropogenic pressures given that they are the receptacle of all the contaminants and nutrients from the upper river watersheds (Howarth et al., 2011), from both point sources (urban and industrial wastewater) and diffuse sources (agriculture)."*

**RC1.2:** Table 1. I am a little confused to see that the Dordogne watershed is included in the third row of the table and not in the second. The same goes for the Nive, I suppose, in the Adour column. In general, this 3rd row (Estuary basin area) should be better explained or its name should be changed.

**AC1.2:** Indeed, in this study, the Dordogne river basin area should be included in the River basin area because its water and nutrient loads also enter the estuarine system as upstream boundary conditions for the C-GEM model. To prevent any confusion, a brief statement was added to the note of Table 1:

*"(2)* River basin area includes the basin area of the rivers directly flowing into the estuary."*

However, the Nive River was not explicitly considered in the study of the Adour estuary, as we couldn't find any data neither for water flux nor nutrient fluxes for this tributary. Therefore, the basin of the Nive River, which represents less than 10% of the total surface area, is not considered in the area of estuary in our table.

**RC1.3:** Section 2.3.1: In this manuscript, the model is a black box and it is necessary to consult the articles of Volta et al. to know the details of the modeled processes. For example, I tried to see how the interactions between SPM and phosphates were accounted for, but I could not find this information. Wouldn't it be appropriate to put the model elements in an appendix?

**AC1.3:** We thank the reviewer for this suggestion. Upon reading the reviewer's comment, we acknowledge that, without explicitly describing the full set of equations governing the model, C-GEM appears as a black box in our manuscript. This was also suggested by other reviewers. We initially only provided references to Volta et al.'s work where all the model description is detailed in order not to dilute the message of our paper too much. Indeed, our study is more a model-wide application and not a new model development. In the updated version of the manuscript, we followed the reviewer's advice and included the equations governing all the biogeochemical state variables in the supplementary material.

We introduced the following statement at the beginning of section 2.3 of the manuscript:

*"The extensive description of the model and its underlying assumptions is available in Volta et al. (2014) and the following sections only briefly describe the state variables and processes included in the biogeochemical module as well as the modifications introduced for the simulations discussed in this manuscript. However, all the equations governing the production and consumption reactions of all state variables as well as their parameterization are provided in the supplementary material."*

Moreover, note that, in the current set-up of C-GEM, the adsorption and desorption of the particulate P on SPM is not included, which is now explicitly stated in the manuscript (see answer **AC1.17**) for further discussions as to what motivated this choice).

**RC1.4:** Line 177. The time resolution of the model is hourly and daily depending the parameter. However, the input data to the model does not have this resolution. Here, it is stated that the data

was linearly interpolated to obtain the temporal resolution of the model. Could you give some indication of the frequency of acquisition of the measurements of the Naiades data set?

**AC1.4:** The calculation time step of C-GEM is relatively short (300 seconds) in order to comply with numerical stability constraints resulting from our advection scheme. This short time step also allows a fine representation of short-scale physical and biogeochemical processes that may occur in estuarine settings (tides or light-dependent biological processes, for instance). The hourly resolution at the marine boundary conditions is used in order to resolve tidal fluctuations and was only possible because another numerical model (ECO-MARS3D) is used to provide such high-resolution coverage. Upstream, however, the resolution of the boundary conditions is dependent on the sampling frequency of the available data. We clarified this in the updated manuscript and, in particular, provide the frequency of the measurements (including the data from NAIADES), which are now presented in section 2.2 Data Collection (in the second paragraph).

*"The temporal resolution of the water quality data acquired is generally monthly or bimonthly. Chl a concentrations were usually available only from March to September (main phytoplankton growth period)."*

**RC1.5:** Table 3: concentrations unit in µmol/L, and in Table 4, units in mg/L

**AC1.5:** The unit used in the model for its calculations is µmol/L while most of the collected data are in mg/L.

In this paper, simulations from the model were thus converted into the unit of mg/L to facilitate the comparison with measured data (in unit mg/L).

See also **AC2.3:** In general, µmol/L is frequently used in marine systems sciences while mg/L is more used in river systems investigations. Whereas, the calculations are made in moles in the model, in this study we used the mg/L (N, P, Si, C) instead of µmol/L to facilitate comparison with the observed data all in mg/L for the import from the rivers and within the estuaries. Also, we considered that the calculation of material fluxes is more meaningful when they are expressed with their respective molar mass (14, 31, 28, 12 respectively for N, P, Si, C). However, to reconcile the two disciplines we have added some important values in both units.

**RC1.6:** Table 4: the value of SPM concentration of the Dordogne is very high: it is not representative of the Dordogne river. It is most likely a value from a station located in the tidal estuarine part of the Dordogne river. The Dordogne is a river with many upstream dams: SPM concentrations are low until the tidally influenced zone is reached. It would be interesting for readers familiar with these estuarine systems to give the names of the stations used to define the river mixing end-members.

**AC1.6:** The Dordogne River in this study is considered as an input to the estuary with a daily time step in terms of water and biogeochemical elements, differently from the upstream of the Gironde (Garonne River) which is influenced by tidal cycles. The values used for the Dordogne river come from the closest sampling station to the confluence point (station number: 5026000) which is indeed influenced by the tide. The high annual mean value was caused by the SPM measured during Aug-Oct (for example, 7900 mg/L in Aug 2015, 3600mg/L in Oct 2015,

2400mg/L in Sept 2016) while in March the values were only 10-29 mg/L. The name and location of the station used as the boundary condition for the model for the Dordogne are now provided in the manuscript with a word of caution regarding the SPM concentrations at the end of section 2.3.3:

*"The closest available sampling station (number: 5026000, from NAIADES) on the Dordogne river is around 32 km to the confluence."*

*"The Dordogne River was considered as a source of biogeochemical elements for the Gironde estuary at the confluence. This ignored the tidal cycle effects on the tributary, and might cause deviations downstream the confluence."*

Please see also **AC1.8**.

**RC1.7:** Page 12 and Fig 3. The model results are compared to values measured along the estuaries at two contrasting dates, one during a period of high tributary discharge and the other during a period of low discharge. However, one date, for example January 15, 2015, is a 24-hour period when there were 2 tidal cycles. For these macrotidal estuaries, it is likely that the timing of the tide plays an major role in the spatial distribution of the different compounds described here. However, I did not see when this tidal cycle effect was discussed. Do the model outputs correspond to a time of high tide? low tide? For an estuary the size of the Gironde, a given time corresponds to different moments in the tidal cycle upstream and downstream. This would need to be discussed.

**AC1.7:** We agree with the reviewer that, in estuarine systems, particularly in the most downstream section, the time of measurement may significantly affect the value of any given variable because of the tidal influence. Our model, with a calculation time step of the order of a few minutes (300 seconds), and hourly marine boundary conditions, perfectly resolves the tidal cycle. This was abundantly demonstrated in previous publications using C-GEM (Nguyen et al., 2020; Laruelle et al., 2019; Volta et al., 2016). Furthermore, the ability of C-GEM to capture tidal variations in the Seine estuary was demonstrated in Laruelle et al., 2019 with a similar set-up and transient simulations.

In our three years long simulations, the temporal resolution of the model outputs is set at 4 hours. This value was selected in order to limit the size of the output files but nonetheless provides most of the amplitude of the variation in concentration of the state variables of the model over a tidal cycle. In Figure 3, 4, S-1, S-2 and S-3, we thus do not indicate a single value for the different variables but an envelope that represents the amplitude of the variations over two tidal cycles for the considered date. The envelope consists of the maximum and minimum values over a span of 24 hours at a temporal resolution of 4 hours. For instance, significant tidal variations can be observed in the Charente for most variables in its downstream section. The temporal resolution of the model calculations and as well as that of its results files and the ability of the model to fully resolve tidal variations are explicitly stated in section 2.3.5 of the manuscript:

*"The relatively short residence time, combined with hourly boundary conditions at the mouth of the estuary allows an accurate resolution of the tidal cycle in the estuaries, including during transient simulations as was demonstrated by Laruelle et al. (2019). The time resolution of the model outputs was set at 4 hours in order to minimize the size of the export files while capturing most of the tidal and diurnal variability. In the figures there-after, the envelope around the model*

*results represents the minimum and maximum values over two tidal cycles in order to provide the amplitude of the tidal influence on concentrations at different locations of the estuary."*

**RC1.8:** Line 295 "…downstream of the confluence with the Dordogne tributary". This sentence implies that the Dordogne is a source of material for the Gironde estuary at the confluence. This ignores the reality of the environment. The Dordogne and the Garonne meet in the estuarine zone. At the confluence, the waters of the two rivers are very efficiently mixed by the tide: mixed waters of the Garonne and the Dordogne rise largely upstream of the confluence, up to the dynamic tidal limit, located on both rivers more than 70 km from the confluence (e.g. Parra et al., Continent Shelf Res 19, 135-150, 1999)

**AC1.8:** In this study, Dordogne was simplified as a source of material for the Gironde estuary at the confluence, which was the easiest way to set up the model for this study. Indeed, the purpose of the paper was to test the performance of a 1D model with simple inputs and settings on a variety of estuaries, e.g. with different geomorphologies, rather than focusing on one specific estuary, which can be done locally with more available data as the model is open source. The results presented in this paper showed that even with this simplification of the Dordogne River, the biogeochemical processes were well represented both along the estuary and on specific cross-sections.

Words of cautions were added in discussion (4.1 Model Applicability and Limitations):

*"The Dordogne River was considered as a source of biogeochemical elements for the Gironde estuary at the confluence. This ignored the tidal cycle effects on the tributary, and might cause deviations downstream the confluence."*

Please see also **AC1.6**.

**RC1.9:** For some parameters shown in Figure 3 (and Fig S1), only the data from the river mixing end-member and the data from the marine mixing end-member are shown. There is no data in the estuarine part: we can therefore not talk about model validation here (e.g. the bottom graphs for the Loire). For the Adour, there is only one control point. Is it a point taken at high tide or low tide?

**AC1.9:** Figure 3 and Figure S-1 were dedicated to present the calibration results for 2015. Two specific dates were presented. In Figure 3 and Figure S-1, we presented all the data we could get for those dates. Monitoring along the estuaries was not carried out as frequently as on rivers, or/and the sampling data for estuaries are not accessible. Therefore, in Figure 3 and Figure S-1, there were no data for some variables along some estuaries, such as TOC, DO and Chl a for the Loire.

The validation was carried out through the whole 2014 and 2016 years, based on seasonal available data for the middle of the estuaries in Figure 4. Besides, through other estuaries where there were data, we can see that the model represents the longitudinal variations. Indeed, along the Adour estuary, there was only one station (station number: 5200200, from NAIADES) at 22 km within the estuary (Figure 3). However, the observations for this station fit rather well with the seasonal simulations for the period 2014-2015. Further, in Table 7, all the observations and corresponding simulations are taken into account for the evaluation of the model's performance.

Again, our approach was to analyze a variety of estuaries with the existing data aiming at pushing towards additional field investigations where no data exist on the French Atlantic coast but also for >90% of the world. We want to restate that one of the aims of our study is to provide new insight on smaller, seldom studied estuaries rather than exploring the few already well-known systems that only represent a fraction of the Atlantic French coastline. Besides, although a dedicated 3D model is obviously insightful for the rare systems where sufficient data are available, a simplified 1D model can be useful for exploring some hypotheses.

**RC1.10:** Generally speaking, most of the control points are located either at the level of the marine end-member or in the zone where the salinity is close to zero. There are not many control points in the salinity gradient area. This makes it difficult to claim that the model calibration is robust.

**AC1.10:** It is true for some estuaries, but for large estuaries, calibration/validation points (sampling stations) are also located along the estuaries, besides for both marine and riverine boundaries, except for some variables. Medium and small estuaries are shorter, and thus have fewer sampling points along the estuaries. We acknowledge that this can be a caveat for small estuaries, which can have an important impact on their adjacent coastal zone.

Validation stations far from the riverine boundary were added in the supplementary material (Figure S-3) to better show the performance of the model (see also below **AC1.12**).

We would also like to stress out that we performed an extensive data search to gather as much estuarine data as we could for all the modeled systems. The scarcity of sampling locations within the salinity gradient of smaller estuaries is obviously a hurdle in the way of a better understanding of the biogeochemical cycling of nutrients within these systems. However, we argue that ignoring such systems and leaving them out of modelling investigations until their data coverage improves, is not ideal and provides a skewed perception of the role of estuaries at the national scale. Rather, we advocate for a modeling strategy that relies on a simpler, yet robust and proven model with limited data demand, that allows regional application where data-rich and -poor systems can be investigated at the same time in order to progress toward a more representative regional picture of the role of estuaries as a filter between French watershed and coastal Atlantic waters.

**RC1.11:** Line 344: data expressed in % saturation would allow a more direct visualization of O2 consumption

**AC1.11:** Following the reviewer's suggestion, we now provide in the text some % saturation values between brackets in some places to help the reader interpret our $O_2$ concentrations.

**RC1.12:** Figure 4. Simulated and measurement of salinity are important information that is missing here. Indeed, it is stated in line 316 that the stations were chosen in such a way that they are not too influenced by the marine boundary. But these stations should not be strongly influenced by the river boundary as well. Indeed, according to figure 3, all stations in figure 4 are in the area where the salinity is close to 0.

Here again the question of the tide comes up: for the selected stations what is the variation of the concentrations during a tidal cycle in summer, in winter, during spring tides or neap tides? This aspect is not discussed here. Measurements on tidal cycles probably do not exist, but can the model simulate them? If not, this role of the tide should still be discussed.

**AC1.12:** Salinity were added in Figure 4 and S-2.

Validation stations far from the riverine boundary were added in the supplementary material (Figure S-3) to better show the performance of the model.

Again, we understand the valid concern of the reviewer regarding the potential effect of tides on the results and acknowledge that it was an oversight on our part not to explicitly state that the model does capture tides (see answer **AC1.7**) and how we tried to represent tidal variations onto our figures by representing the minimum and maximum values over two tidal cycles onto our graphs. These do provide an estimate of the tidal influence on the concentrations simulated by the model in the most downstream sections of the estuaries.

**RC1.13:** Section 3.2.1: in this paragraph I found it difficult to know whether reference was made to flows from estuaries or river flows into estuaries.

**AC1.13**: In order to be more explicit, "ingoing" and "outgoing" were changed to "import" (to the estuary) and "export" (from the estuary), respectively, throughout the manuscript.

**RC1.14:** In line 361 and in table 8, the flow results are given with 2 decimal places. Is this level of precision justified?

**AC1.14:** We agree that the two decimals are not justified and they were modified accordingly to only 1 decimal place.

**RC1.15:** Line 373: "They accounted for about 80% of the total water discharge from all the estuaries on the French Atlantic coast and 83% of the total watershed areas on the French Atlantic coast, and hence a similar proportion of the nutrient fluxes.": The missing 20% are represented by small rivers with small estuaries. As a result, this 20% certainly has a lower nutrient retention rate than the average of the estuaries studied here. Thus, the contribution to the nutrient flow to the Atlantic coast of this 20% is probably higher than 20%.

**AC1.15:** You are right about this. The sentence has been changed as follows:

"*They accounted for about 80% of the total water discharge from all the estuaries on the French Atlantic coast (based on long-term analysis of runoff data from French national databases) and 83% of the total watershed areas on the French Atlantic coast. However, under the assumption (supported by our simulations and previous literature, e.g. Nixon et al., 1995) that smaller estuaries such as those not considered in our study, likely have smaller retention rates, these 7 estuaries might produce nutrient fluxes slightly lower than the 80% of the total water fluxes of the French Atlantic coast.*"

**RC1.16:** Are the nutrient import data in Table 8 and Figure 7 direct outputs from C-GEM? It is not clear to me

**AC1.16:** Yes, they are. In the first paragraph of section 3.2.1, it pointed out that the values in Table 8 were calculated from model. Figure 7 used the values calculated from the fluxes which were calculated from the model.

RC1.17: Table 7: TP values do not take into account the P associated with inorganic particles. However, some of this P is desorbed in the salinity gradient, so that an estuary can become a source of DIP (e.g. Deborde et al. 2007 L&O 52, 862-872). Is this reaction taken into account in the model?

**AC1.17:** In its current state, the model does not take into account this process. The limitations of the model associated with its current level of complexity (i.e. lack of explicit representation of the sorption/desorption mechanism for P or the crude representation of the interaction with the sediment) are now discussed in the second paragraph in section 4.1, which was entirely rewritten. This new section also includes justification for our choice to use a relatively simple biogeochemical module in our simulations. Further considerations about the matter are also available in answer **AC3.9**. We thank the reviewer for the reference provided that now is cited in our manuscript.

"*In its current setup, the biogeochemical module of C-GEM considers some of the most essential biogeochemical processes and reactions (i.e. primary production, organic matter degradation, denitrification…). In spite of generally good ability of the model to capture the main spatial and temporal biogeochemical dynamics of the different systems studied (i.e. longitudinal, seasonal and amplitude of the variations of nutrients carbon and oxygen fields), several potentially important processes contributing to the N and P cycling in estuarine environments in particular are still ignored or largely simplified. These include benthic-pelagic exchanges, sorption–desorption of phosphorus, mineral precipitation or a more complex representation of the biological planktonic/benthic compartments (such as grazing by higher trophic levels, or multiple reactive organic carbon pools for instance). This limits the depth of mechanistic understanding that the model can provide of nutrient cycling, particularly regarding interactions between pelagic and benthic compartments which can significantly influence the intensity but also the timing of nutrient and organic matter cycling in estuaries (Laruelle et al., 2009). The addition of a full diagenetic module at each grid cell of our model would be possible but would also increase its calculation time by one order of magnitude and require a very long spin-up to generate initial conditions for the benthic species. There exist simpler benthic modules of lower complexity, which would limit the computation cost of adding an explicit representation of benthic processes (see Soetaert et al., 2005) but those would nonetheless significantly increase the data demand of the model to be properly calibrated. Thus, while we believe the inclusion of an explicit benthic compartment to our model is the way forward on the long run, such an increase in complexity without sufficient data for a proper calibration and evaluation might introduce more uncertainty than actual mechanistic understanding to the model. In the present study, a simple representation of particulate matter burial was nonetheless implemented and applied to phytoplankton and TOC to provide a first-order representation of the process, which is necessary to evaluate the retention of carbon and nutrients within the system. We believe this addition, coupled with denitrification provides a first insight on the main pathways removing nutrients from estuaries.*"

**RC1.18:** Fig. 7: This figure shows average calculations of retention rates and residence times at the annual scale. For me, an annual average is meaningless and does not explain the relationship between the two parameters. It would be more interesting to compare these two properties in flood and low water periods. The relationship should certainly be better and the processes that go with it should be easier to explain.

**AC1.18:** Thanks for your suggestion. Figure 7 presented firstly on an annual average was to show that generally, the larger residence time caused a larger retention rate. According to your suggestion, we added a figure of average calculations of retention rates and residence times for the summer season (May-Oct) which further showed that the estuaries had larger residence time during the dry season, which also led to larger retention rates.

The text was modified accordingly:

*"We also calculated retention rates and residence times for summer season (May-Oct) which further showed that the estuaries had larger residence time during the dry season, which also led to larger retention rates. The linearity of the relationship is not so well adapted, % retention plateauing at high residence times."*

[Figure]

Figure 1 The relations between the annual retention rate for nutrients (total nitrogen (TN), total phosphorus (TP), total silica (TSi), and total organic carbon (TOC)) and the fresh water residence time for the estuaries studied (the Somme, Seine, Vilaine, Loire, Charente, Gironde, and Adour estuaries) for the 3 years 2014–2016. (a). Annual mean; (b). Summer season (May-October) mean.

**RC1.19:** Line 472: simulations do not resolve the tidal cycles.

**AC1.19.** We apologized again for not being clear enough on this issue in the manuscript. The model does totally resolve tidal cycles for hydrology and biogeochemistry. Please see answers AC1.7 and AC1.12 for further information on the matter.

**RC1.20:** Line 481 to 486: I have the impression that we are going in circles in this paragraph.

**AC1.20:** The sentence was updated

*"Further, solid results were gained elsewhere with C-GEM supporting its genericity. i.e. carbon processing in the six major tidal estuaries (length >80 km) flowing into the North Sea (Volta et al., 2016b), biogeochemical dynamics and $CO_2$ exchange in three tidal estuaries (length >90 km, Volta et al. 2016a), $CO_2$ evasion on 42 tidal estuaries along the US east coast (Laruelle et al., 2017) and the Seine (Laruelle et al., 2019), biogeochemical processes and fluxes on a tropical estuary (Nguyen et al., 2021)."*

**RC1.21:** Line 503: perhaps it should be recalled here that the link between water residence time and nutrient retention is a known phenomenon for lakes, wetlands or dams and that it is this principle that leads to the restoration or construction of wetlands

**AC1.21:** A sentence has been added to address this question.

*"The role of residence time on nutrient retention/elimination is not only the fate of estuaries but also of rivers, stagnant systems (lakes, ponds and reservoirs), and also wetlands. This function can be valued for reducing contaminations and sometimes even promoted for restauration although the best way is to limit the amount of fertilizer added (Bernot and Dood, 2005)."*

**RC1.22:** The first paragraph of section 4.3 should be included in the introduction

**AC1.22:** Thanks for this suggestion, and the correction was made in the first paragraph of the Introduction.

I hope that my comments and considerations will make it possible to better highlight the quality of the results of this modelling, which has interested me greatly.

We greatly acknowledge Prof. Pierre ANSCHUTZ for his constructive remarks based on a large view on estuaries, specifically here those of the French Atlantic coast. We indeed believe that his insightful comments helped us improve our manuscript.

We greatly thank all the reviewers for their thorough and helpful comments which contribute to improving the manuscript.

Please find our point-by-point responses (in blue) in the following, and changes in the manuscript are *in italic* here.

**RC2: Anonymous Referee #2**

In this manuscript Wei et al apply an existing model to 7 macrotidal estuaries along the French coast for the years 2014 – 2016 to simulate impacts of estuarine characteristics on riverine nutrient fluxes to the coastal Atlantic. The model used has been tested widely across different systems and modeled parameters compare well with observed values. The paper is well written, and the main conclusions that large estuaries have higher retention rates presumably due to higher residence times is clearly supported by the results.

In my view, two aspects need attention: 1) Riverine particulate (and organically bound) nutrient input and 2) impact of import particulate (organic) matter from the coastal Atlantic.

**RC2.1:** L85ff: Estuarine circulation is driving the accumulation of marine particulate matter and instrumental in the formation of the Turbidity Maximum (Burchard et al., 2018). How is this solved in a 1-D model? Import of marine organic matter can be an important source of nutrients in the estuary and how is this accounted for in your model set-up?

**AC2.1:** We agree that estuarine circulation controls particulate matter accumulation and is thus instrumental in the formation of the Turbidity Maximum. However, the work of Savenije (2012) on the interplay between estuarine geometry and hydrodynamics in alluvial estuaries, on which relies C-GEM's physical model, demonstrated for a large number of systems that the main hydrological properties of an alluvial estuary can be represented by an idealized geometry that can itself be modeled as a longitudinal 1-dimension, depth and width varying model. In C-GEM, the SPM dynamics are controlled by erosion and resuspension processes in addition to the advection and dispersion to which are submitted all variables within the water column as is the case with any reactive transport model, regardless of its number of spatial dimensions. This formulation allows, at each time step and each grid cell, to account for the local effect of hydrodynamics on suspended material and the influence of the concentrations upstream and downstream. This also includes the effect of the marine boundary condition, which can as pointed out by the reviewer partly control the SPM dynamics within an estuary. We acknowledge however that our model only included a single pool of SPM and thus does not differentiate marine from riverine suspended material.

While the complete description of our SPM module is available in Volta et al. (2014) which is referred to in our manuscript, we agree that providing the equations governing the SPM dynamics in the model will be useful to the reader and we thus introduced these equations in the supplementary material.

While we believe that the ability of our model to capture the location of the turbidity maximum in each system as well as a realistic range of values for SPM concentrations is satisfying in the context of our modeling exercise, we also acknowledge that a fine simulation of SPM dynamics

in estuarine setting requires a dedicated model with intensive calibration and that even a fully blown 3D model will not necessarily be able to achieve such task will limited data.

A paragraph was however added in the manuscript to better explain this issue in the model presentation.

*"The SPM dynamics is controlled by transport of suspended material (i.e. advection and dispersion) as well as local deposition and resuspension/erosion processes but the model does not distinguish between the pools of marine and riverine suspended material. P adsorption and desorption to particulate material to form an iron-bound complex for example is not accounted for. Thus, the only control exerted by SPM concentrations in the water column on the other biogeochemical variables occurs through the influence of SPM on the light extinction coefficient, which partly controls primary production."*

**RC2.2:** L129ff: The data used include inorganic nutrients. However, for a nutrient budget it is important to estimate total nutrient loads as many nutrients may be bound to organic matter (either dissolved or particulate and in case of P, Fe-bound PO4 may be important). Especially in case of riverine phytoplankton blooms, particulate loads may be significant. Living phytoplankton may only capture a small part of the total particulate nutrient load (e.g.Hillebrand et al., 2018) due to a substantial fraction of detritus. How do you account for this?

**AC2.2:** We understand the concern of the reviewer and acknowledge (as is the case for SPM dynamics, see answer above) that not including the equations governing the consumption and production rates of the different state variables of the model make it a bit difficult to fully understand what is and what is not accounted for in the budgets presented in the manuscript. The particle-bound nutrients associated to inorganic compounds such as Fe-bound P are not considered in the model and a sentence was added in the description of the model to make this clear.

*"In its current setup, the biogeochemical module of C-GEM considers some of the most essential biogeochemical processes and reactions (i.e. primary production, organic matter degradation, denitrification…). In spite of generally good ability of the model to capture the main spatial and temporal biogeochemical dynamics of the different systems studied (i.e. longitudinal, seasonal and amplitude of the variations of nutrients carbon and oxygen fields), several potentially important processes contributing to the N and P cycling in estuarine environments in particular are still ignored or largely simplified. These include benthic-pelagic exchanges, sorption– desorption of phosphorus, mineral precipitation or a more complex representation of the biological planktonic/benthic compartments (such as grazing by higher trophic levels, or multiple reactive organic carbon pools for instance). This limits the depth of mechanistic understanding that the model can provide of nutrient cycling, particularly regarding interactions between pelagic and benthic compartments which can significantly influence the intensity but also the timing of nutrient and organic matter cycling in estuaries (Laruelle et al., 2009). The addition of a full diagenetic module at each grid cell of our model would be possible but would also increase its calculation time by one order of magnitude and require a very long spin-up to generate initial conditions for the benthic species. There exist simpler benthic modules of lower complexity, which would limit the computation cost of adding an explicit representation of benthic processes (see Soetaert et al., 2005) but those would nonetheless significantly increase the data demand of*

*the model to be properly calibrated. Thus, while we believe the inclusion of an explicit benthic compartment to our model is the way forward on the long run, such an increase in complexity without sufficient data for a proper calibration and evaluation might introduce more uncertainty than actual mechanistic understanding to the model. In the present study, a simple representation of particulate matter burial was nonetheless implemented and applied to phytoplankton and TOC to provide a first-order representation of the process, which is necessary to evaluate the retention of carbon and nutrients within the system. We believe this addition, coupled with denitrification provides a first insight on the main pathways removing nutrients from estuaries.*"

However, N and P associated to dead phytoplankton and detritus organic matter is indeed considered in the model. We assume Redfield ratios for this organic matter pool based on TOC and the corresponding organic N and P are included in our subsequent calculations and are thus part of our TN and TP pools. In the model, when TOC is remineralized, DIN and DIP are released following the Redfield ratio. This is now also mentioned in the model description and as we understand those are fundamental information to have in order to understand how our budgets are calculated:

"*Note that the concentrations of the organic state variables (Dia, nDia and TOC) are expressed in µmol of carbon per liter but the model uses Redfield ratios to account for the associated amounts of N, P and, in the case of diatoms, Si. Thus, the variable TOC actually includes all detritus and is sustained by the death of phytoplankton and its aerobic degradation fuels the stocks of dissolved inorganic nutrients.*"

Overall, we made an effort in the updated manuscript to better explicit what the different pools of N and P correspond to. It was also pointed out by reviewer 1 (**AC 1.1**) that our model may look like a black box because of its limited mechanistic description and we want to restate that this was not intentional. We tried to make our model description more transparent in the updated version of the manuscript and further discuss the potential implication of the current level of complexity of our SPM and biogeochemical modules in section 4.1, (see answers **AC2.9**, **AC3.3** and **AC3.9**, for further considerations on the matter).

Please see also **AC1.17**, **AC2.1**, **AC2.2**, **AC2.11** and **AC2.12, AC3.9** for new inputs on N,P cycling and TP:TN ratio.

**RC2.3:** Furthermore, I suggest to use moles throughout the text.

**AC2.3:** In general, µmol/L is frequently used in marine systems sciences while mg/L is more used in river systems investigations. Whereas, the calculations are made in moles in the model, in this study we used the mg/L (N, P, Si, C) instead of µmol/L to facilitate comparison with the observed data all in mg/L for the import from the rivers and within the estuaries. Also, we considered that the calculation of material fluxes is more meaningful when they are expressed with their respective molar mass (14, 31, 28, 12 respectively for N, P, Si, C). However, to reconcile the two disciplines we have added some important values in both units.

Please see also **AC1.5**.

Further smaller comments:

**RC2.4:** L 67-70: This sentence is difficult to understand (…despite mixing curves have (having?) been useful… Furthermore, I suggest to elaborate a bit on the limitation of using mixing curves. Wouldn't they have been useful to constraint he presented budgets?

**AC2.4:** Thanks for the suggestion, and a new sentence was added.

"*Also, mixing curves are meaningful when water quality data are numerous within the salinity gradient, which is not possible for many estuaries.*"

In fact, as far as possible we used the observations available along the salinity gradient and compared them with the model simulations, which is another way to use these mixing curves. However, while mixing curves allow to tell qualitatively about the source or sink of a given element within this gradient, the modelling approach allows a quantification.

**RC2.5:** L185ff: The Dordogne has exceptionally high SPM (>40 * other rivers). What is the reason and what is the impact of this on the model outcome of the Gironde?

**AC2.5:** The SPM values used for the Dordogne River come from the closest gauging station to the confluence point (~33 km upstream to the confluence) which is influenced by the tide. High annual mean value was caused by the SPM measured during Aug-Oct (for example, 7900 mg/L in Aug 2015, 3600mg/L in Oct 2015, 2400mg/L in Sep 2016) while in March the values were only 10-29 mg/L.

Please see **AC1.6**.

"*The Dordogne River was considered as a source of biogeochemical elements for the Gironde estuary at the confluence. This ignored the tidal cycle effects on the tributary, and might cause deviations downstream the confluence.*"

**RC2.6:** L212ff: How would an import of SPM from the coastal Atlantic influence the parameters listed in Table 5? By which modelled processes is the SPM-max generated?

**AC2.6:** Please refer to answers **AC2.1** and **AC2.2** for a more detailed comment about our SPM module but our model both takes into account local processes of SPM erosion and deposition as well as transport through advection and diffusion. Thus, the influence of high (or low) SPM concentrations at the marine boundary condition of the model will influence the upstream profile of SPM. This was made possible by the hourly temporal resolution of the MARS 3D model that provided boundary conditions for our simulations.

**RC2.7:** L240: Again, how is dealt with particulate/organically bound nutrients?

**AC2.7:** The particle-bound nutrients are not considered in the model, however, the detritus N and P pools are explicitly considered. Please see answers to questions **AC2.1** and **AC2.2** for more clarifications on the matter and **AC3.9** for a justification regarding the level of complexity of the model we used.

**RC2.8:** L273ff: Are the high Gironde values for SPM driven by the high Dordogne values? Is the SPM in the SPM max mainly riverine or marine?

**AC2.8:** Unfortunately, the model does not explicitly distinguish riverine SPM from marine SPM. However, the extreme SPM values simulated around 90 km do indeed correspond to the confluence with the Dordogne. They are thus likely influenced by the particularly high SPM concentration reported in the latter. The actual SPM concentrations within the Dordogne waters when they mix with those of the Gironde are likely lower than those used in our simulations (which were the closest available ones). This possibly results in an overestimation of this SPM peak relatively far upstream of the Gironde estuary. Following reviewer 1 (See **AC1.6**), a word of caution was introduced in the model description and an additional comment was introduced in the discussion of the results.

**RC2.9:** L285ff: Desorption of PO4 is mentioned but from the model description I notice that this is not implemented in the present model. Please explain.

**AC2.9:** Adsorption and desorption are indeed not considered in our model. As a consequence, we should not have referred to this process when describing our model results. The corresponding sentence has thus been removed from the text and the potential implication of omitting this process is now discussed in section 4.1 of our manuscript. The potential influence of the lack desorption of PO4 in our model, as well as the justification for its current level of complexity is now also further discussed in the second paragraph of section 4.1 of the manuscript on the limitations of the model. Furthermore, these considerations have also been partly discussed in answers to comments from the other reviewers (see answers **AC2.2**, **AC3.3** and **AC3.9**).

**RC2.10:** L289: I suggest to use moles instead of grams to indicate nutrient concentrations. Please note that phosphorus is misspelled as phosphorous at several instances throughout the text and in tables.

**AC2.10:** Please see **AC.1.5** and **AC2.3** for the use of moles and grams.

In general, µmol/L is frequently used in marine systems sciences while mg/L is more used in river systems investigations. Whereas, the calculations are made in moles in the model, in this study we used the mg/L (N, P, Si, C) instead of µmol/L to facilitate comparison with the observed data all in mg/L for the import from the rivers and within the estuaries. Also, we considered that the calculation of material fluxes is more meaningful when they are expressed with their respective molar mass (14, 31, 28, 12 respectively for N, P, Si, C). However, to reconcile the two disciplines we have added some important values in both units

Spelling mistakes have been corrected in the paper.

**RC2.11:** L361ff: If riverine TN is derived from DIN + N in phytoplankton, the TN load may be underestimated as riverine particulate matter may consist of phytoplankton detritus (see e.g. Hillebrand et al., 2018). Also DON and DOP are not accounted for. The potential underestimation of TN and TP loads should be discussed.

**AC2.11:** This fundamental issue regarding how we calculate TN and TP has been clarified in answer **AC2.2** and the manuscript has been updated in several places to make this clearer. In the model, TN and TP correspond to the N and P associated with dissolved nutrients + phytoplankton + TOC, which essentially correspond to detritus organic matter. The latter two pools of N and P are calculated from the C content of the Dia, nDia and TOC state variables of the model using Redfield ratios. Please see the added paragraph in the answer **AC2.2** and the discussion of the drivers of the TN/TP ratio in **AC2.12** below.

**RC2.12:** L381ff: What drives the differences in TN/TP ratio?

**AC2.12:** In our simulations, because phytoplankton and organic matter are assumed to follow the Redfield ratio, burial and denitrification control the TN/TP ratio along the estuary as well as, to a degree, the concentrations at both boundaries. For instance, denitrification will be a loss of N from the system and thus will affect the TN:TP ratio. In the case of burial, it is assumed that the material buried (phytoplankton and organic matter) has a fixed (Redfield) N:P ratio thus removing the same proportion of N and P. However, the ratio of inorganic (DIN or DIP) to organic pools is different for N and P. The net effect of burial thus also affects the TN:TP ratio of the system. This notion has now been introduced into the manuscript.

*"Overall, TN and TP retention rates are comparable in the estuaries simulated, but some system are more efficient at removing P while other are more efficient at removing N. This implies that the TN/TP ratio varies along the estuarine gradient. These variations are controlled by the complex interplay of phytoplankton and organic matter (TOC) burial and denitrification along the estuary. Denitrification, being a net removal of N obviously increases the TN:TP ratio. However, the effect of phytoplankton and organic matter burial is more subtle. In the model, it is assumed that the material buried (phytoplankton and organic matter) has a fixed (Redfield) N:P ratio and thus removes the same proportion of N and P through burial. However, the ratio of inorganic (DIN or DIP) to organic matter (associated with phytoplankton or TOC) is different for N and P. As a consequence, the net effect of burial also affects the TN:TP ratio of the system."*

**RC2.13:** L441: Unclear sentence: …often unavailable data sets (needed for?)

**AC2.13:** We better specified for often unavailable data sets as follows:

*"The 1-D biogeochemical model C-GEM was built to overcome the requirement of large, often unavailable data sets needed (e.g. geometry at a fine resolution) for implementing complex multidimensional models…"*

**RC2.14:** L507: which elimination process could be responsible (other than denitrification and sedimentation?)

**AC2.14:** Denitrification is the main elimination process that consumes $NO_3$ while nitrification is the main elimination process transforming $NH_4$.

**RC2.15:** L524: what is meant by diatom outburst under osmotic pressure?

**AC2.15:** Freshwater diatoms can break out when entering in the salinity gradient. The formulation has been changed:

*"TSi was eliminated only slightly (e.g., 2% for the Seine estuary in 2016 and the Loire in 2015 and 1% for the Adour in 2014 and 2015), suggesting that no diatom uptake occurred or was compensated by freshwater diatom mortality sensitive to the salinity gradients (Ragueneau et al., 2002; Roubeix et al., 2008)."*

Literature cited

Burchard et al. (2018) Sediment Trapping in Estuaries. Annual Review of Marine Science. Vol. 10:371-395

Hillebrand et al., (2018) Dynamics of total suspended matter and phytoplankton loads in the river Elbe Journal of Soils and Sediments (2018) 18:3104–3113

Thank you for these references that we cited in the text and added to the reference list. Many thanks also for the questions/comments of R#2, which help us to be more accurate on what is accounted in the model or not.

We greatly thank all the reviewers for their thorough and helpful comments which contribute to improving the manuscript.

Please find our point-by-point responses (in blue) in the following, and changes in the manuscript are *in italic* here.

**RC3: Anonymous Referee #3**

The manuscript describes a model that simulates transport and biogeochemistry in various estuaries in France. The 1-D grid is oriented along the flow path, which is constrained at the marine side by results from an ocean model and at the upstream river side by time-series data. The calibration of the model parameterization is based on data from 1 year and then validated based on the ability to reproduce data from the previous and following year. For most locations the model captures the trend of various biogeochemical parameters well, despite major simplifications, which include 1) a 1-D grid that cannot account for the depth-dependency of biogeochemical processes and cannot fully resolve the spatial variability in residence times; and 2) the biogeochemistry in sediments is ignored (it only accounts for organic matter burial).

**RC3.1:** My major criticism is that the role of biogeochemical processes is not well teased out. The reaction rates that the model explicitly resolves are not shown in any figure and the paper would benefit from a more rigorous comparison of the simulated rates with those reported in literature.

**AC3.1:** We understand the comment of the reviewer, which was also expressed by the other reviewers (see answers **AC1.3, AC2.1, AC2.6**). While we carefully referenced the work of Volta et al. in which an extensive description of the model is provided, we acknowledge we did not provide enough information in the present manuscript for the reader to get a good feel for the processes that are and are not included in the model nor how they are formulated. We added in the supplementary material the complete list of the processes (Table S-1) and their formulations as well as the equation controlling the production and consumption of all the state variables included in the model. We believe will help the reader interpret our results with full knowledge of the inner workings of our biogeochemical module.

**RC3.2:** It is unclear to what extent the trends in the modeled output are driven by biogeochemical processes and to what extent by the constantly changing boundary conditions. The influence of the boundary conditions could be especially large for sites where the location for model calibration/validation is near the inlet or outlet (such as in smaller estuaries). The reported retention rates are often low, meaning that the influx and outflux are nearly equal. This could indicate that the respective chemicals do not react much (making model results more trivial), but there could also be a dynamic balance between sinks and sources. The authors may want to improve the manuscript by describing in more detail the cycling of elements within the model domain. A sensitivity analysis would be useful to show to which biogeochemical rate constants the model results are most sensitive. Additionally, a simulation with the biogeochemical reactions turned off would be informative. Without these analyses it is hard to assess if the model captures the biogeochemical dynamics adequately.

**AC3.2:** We understand the reviewer's concern regarding the ability of our simulations to accurately calculate the retention within estuaries because of the potential influence of boundary conditions on our results. The numerical scheme of C-GEM and its short integration time step (300 s) allows for accurate calculations of the lateral fluxes of all state variables between each grid point. It would like to stress out that, while the temporal resolution of the model's output is 4 hours, the calculation of the cumulative transport fluxes are not obtained by the post-processing of those result files but are updated every 300 seconds in order to ensure mass conservation (which was checked prior to performing our simulations). We are thus confident that the exchange at both boundaries, which we use to calculate our nutrients and carbon budgets are accurate even if the retentions we obtain are only of the order of a few percent in estuaries with short residence times. We would also like to stress out that the length of the simulations (3 years, which roughly equals 2000 tidal cycles) prevent transient effects associated with tidally induced changes in concentrations to skew the nutrients and carbon budgets.

Finally, in the updated version of the manuscript, following a request from reviewer 1 (see **AC1.18**), we calculated the retentions rates for nutrients and carbon during the dry season (i.e. Figure 7 and associated text) for the different estuaries investigated. These retentions calculated over a shorter period with lower discharges yield significantly larger retentions and thus illustrate the ability of the model to account for the effect of biogeochemical processing on nutrients and carbon budgets within the systems simulated.

**RC3.3:** In larger estuaries the retention of nutrients is higher, which indicates that biogeochemical processes play a more important role. At these locations early diagenetic processes could also potentially have a larger effect on the water quality. The model contains denitrification in the water-column, but as the water remains oxygenated, it is probably unimportant. Benthic denitrification is likely more important, but not modeled/parameterized. Sediments could also act as a source of nitrogen by releasing DIN derived from remineralized organic matter. The model does neither account for benthic $PO_4$ dynamics nor benthic $O_2$ consumption. The manuscript does not make a compelling argument for why these benthic processes can be ignored.

**AC3.3:** A detailed answer regarding our justification for the lack of a benthic module in the current version of C-GEM is also provided in answer **AC1.18**. We agree that our current set-up does not entirely represent the nutrient dynamics in the location where diagenetic processes may play a large role in N and P cycling. In addition to the justifications provided in answers AC1.17, **AC2.2, AC2.11 &12**, we would also like to mention that, because we performed our calibration, water column processes in C-GEM actually implicitly also account for benthic ones in our calibration. We agree that this is not ideal nor mechanistically accurate but we believe this also limits (for denitrification for instance) the drawback of not explicitly representing benthic processes. Indeed, such a task would require, for the calibration of the new benthic module, additional data such as benthic remineralization rates, denitrification rates that are not necessarily all available in the systems we investigated. Moreover, the addition of a full diagenetic module at each grid cell of our model would increase its calculation time by one order of magnitude and require a very long spin-up to generate initial conditions for the benthic species, which would be difficult to compare with the very scare measurements available. There exist simpler benthic modules of lower complexity, which would limit the computation cost of adding an explicit

representation of benthic processes (see Soetaert et al., 2005 for example) but those nonetheless require more data than we currently have to be applied with a satisfying level of confidence. While we believe this is the way forward in the long run, we would be afraid of adding more uncertainty than actual mechanistic understanding to the model by including too many processes that we cannot properly constrain or evaluate. This rationale to justify the current level of complexity of our biogeochemical model is now further discussed in section 4.1 of the updated manuscript.

*"In its current setup, the biogeochemical module of C-GEM considers some of the most essential biogeochemical processes and reactions (i.e. primary production, organic matter degradation, denitrification…). In spite of generally good ability of the model to capture the main spatial and temporal biogeochemical dynamics of the different systems studied (i.e. longitudinal, seasonal and amplitude of the variations of nutrients carbon and oxygen fields), several potentially important processes contributing to the N and P cycling in estuarine environments in particular are still ignored or largely simplified. These include benthic-pelagic exchanges, sorption– desorption of phosphorus, mineral precipitation or a more complex representation of the biological planktonic/benthic compartments (such as grazing by higher trophic levels, or multiple reactive organic carbon pools for instance). This limits the depth of mechanistic understanding the model can provide of nutrient cycling, particularly regarding interactions between pelagic and benthic compartments which can significantly influence the intensity but also the timing of nutrient and organic matter cycling in estuaries (Laruelle et al., 2009). The addition of a full diagenetic module at each grid cell of our model would be possible but would also increase its calculation time by one order of magnitude and require a very long spin-up to generate initial conditions for the benthic species. There exist simpler benthic modules of lower complexity, which would limit the computation cost of adding an explicit representation of benthic processes (see Soetaert et al., 2005) but those would nonetheless significantly increase the data demand of the model to be properly calibrated. Thus, while we believe the inclusion of an explicit benthic compartment to our model is the way forward on the long run, such an increase in complexity without sufficient data for a proper calibration and evaluation might introduce more uncertainty than actual mechanistic understanding to the model. In the present study, a simple representation of particulate matter burial was nonetheless implemented and applied to phytoplankton and TOC to provide a first-order representation of the process, which is necessary to evaluate the retention of carbon and nutrients within the system. We believe this addition, coupled with denitrification provides a first insight on the main pathways removing nutrients from estuaries."*

Please see also **AC1.17**, **AC2.2**, **AC2.11** and **AC2.12, AC3.9** for new inputs on N,P cycling and TP:TN ratio.

Overall I find the manuscript well written, but it could be further improved. The results section contains many interpretations, which do not belong in this section.

**Specific Comments:**

**RC3.4:** Line 240-246: Sinks and sources in the model are not only related to the inlet and the outlet, but also to exchanges with the sediment and atmosphere. It is specified for Flux_out that it refers only to the outlet. For Flux_in it is stated that it accounts for all inputs. Does this refer only to the river inlet or also inputs from the atmosphere?

**AC3.4:** In our simulations, Flux_in only refer to the river inlet and tributaries. Considering the relatively limited surface areas of the studied estuaries, we do not account for atmospheric deposition of N or P, so the only exchange with atmosphere left would be denitrification, which is considered as part of the retention within the system.

**RC3.5:** The calculated residence time of water is only based on river discharge, but does not account for inputs from the ocean. For the theoretical case that the river influx goes to 0, the water residence time would approach infinity. Obviously, the real residence time of water also depends on the exchange with the ocean and the reported residence times are less meaningful near the ocean boundary. It may be good to point this out.

**AC3.5:** Thanks for this suggestion. We made it clearer by calling it "*fresh water residence time*" in the paper.

**RC3.6:** The estuaries are described to be macrotidal, but I believe the text does not describe the vertical stratification. Perhaps the vertical stratification can be ignored in these macrotidal estuaries. Readers may appreciate a better characterization of the flow in these estuaries to have a better idea of the implications of the 1-D approach.

**AC3.6:** We understand the reviewer's concern and acknowledge that a 1-D model such as C-GEM cannot resolve vertical stratification. However, in macro-tidal estuaries, it can generally be assumed that the water column is well mixed vertically considering the scale of the tidal energy dissipated over relatively shallow water columns. This is reflected in the work by Savenije (2012) for example and we introduced a sentence attesting the overall mixed nature of e.g. the Seine estuary at each tide cycle (Brion et al., 2000) and other of the Atlantic Coast of Europe (Middelburg and Herman, 2007).

*"A 1-D model can be considered as well adapted to these shallow macrotidal estuaries that mixed at each tide cycle as shown by Brion et al. (2000) on the Seine River and Middelburg and Herman (2007) for other estuaries of the Atlantic Coast of Europe, which was also supported by Savenije (2012)."*

**RC3.7:** Line 410-412: The regulation of outflow by the dam is an interesting point. Is this regulation explicitly modeled or averaged over time?

**AC3.7:** This regulation by the dam at the entrance of the Vilaine estuary is not explicitly modelled. The gauging station is located upstream of the dam, so that the model does not take into account the water regulation. Despite this weakness, except SPM, the biogeochemical variables are in a good order of magnitude.

**RC3.8:** Line 451-453: The text states that the grid resolution (2 km) may be too coarse for small estuaries (< 30) km. If the model grid contains less than 15 grid cells, simulations can probably be run quickly. Why did the authors not run simulations with a finer grid resolution?

**AC3.8:** We understand the reviewer's remark. However, our initial aim was to investigate several estuaries with different physical and biogeochemical properties with an identical set-up, both in terms of hydrodynamics (temporal and spatial resolution included) and biogeochemistry (identical reaction rate constants). We thus did not set a finer grid resolution for the small estuaries because we wanted to inter-compare the model performances. We however agree with the reviewer that modifying the grid could be done in future work to improve the model's performances in small systems.

**RC3.9:** Line 457-460: "In the present… denitrification for nitrogen". The text ignores that nutrients are released during remineralization and can be transported back into the overlying water. The statement that denitrification is "the only other potential nutrient removal term" missing in the model is wrong. There are other benthic processes that can act as a sink for nitrogen, sediments can sequester PO4, also Si can be taken up and released by various benthic processes, and there are many other nutrients (e.g. iron) not accounted for in the model. Also, the model does not account at all for the uptake of O2 by the sediment. These benthic processes are often parameterized in ocean circulation models. The benthic remineralization rate could be estimated based on the current burial flux and then be used to approximate the exchange fluxes of O2, P and N. The authors should either implement these processes or explain why these fluxes can be ignored.

**AC3.9:** This is right and we agree that the current level of complexity of our biogeochemical module ignores potentially important processes contributing to the transformations of Si, P and N in estuaries. We do acknowledge these limitations in the text and tried in its updated version to make these statements clearer. We do for instance agree that our statement regarding the burial being the only benthic process able to remove N from the system, other than denitrification is an oversimplification that we paid attention to correct in the updated manuscript.

We are indeed convinced that C-GEM should explicitly include a benthic representation of N, P and Si remineralization as well as O2 uptake. While it is true that such fluxes are included in some models (in particular ocean circulation models as pointed out by the reviewer) and that there exist estimates of some of the reaction rates controlling those fluxes, implementing the latter in our estuarine model would be a massive undertaking which added value remains unclear considering the numerous uncertainties associated with these processes in estuarine settings. Moreover, the implementation of an explicit benthic module may require adding a diagenetic model at each grid cell of the model, which is a massive undertaking. Such a task would require numerous data that do not exist for all the systems we investigated, would largely increase the uncertainty of our results if each new process is not sufficiently constrained, and would likely increase the calculation time by one order of magnitude.

Luckily, there exist alternative solutions with benthic modules of lower complexity and we are working towards the integration of such type of benthic module but those still require ample data collection, and calibration before we can be confident that it would introduce more insight than uncertainty to our results. At current, we believe the current level of complexity of C-GEM already provides valuable insight through this collection of fully transient multi-annual simulations over 7 estuaries. We believe our work is a necessary step towards the implementation of more complex models in the future. We tried to update our manuscript to both make the scope

of our study clearer and better justify the current level of complexity of our model in our simulations.

Please see also our answer **AC3.3**.

**RC3.10:** The fits for the Charente estuary are not so good. Remarkably the trends in simulated PO4 and DO values are opposite to those in the measurements in the calibrated year. Also NH4 concentrations are generally too low. After reading the manuscript it is not clear to me why the model could not reproduce these trends. If the model fails to reproduce the data, should the results be presented in the manuscript?

**AC3.10:** It is true that the model didn't catch the peak and bottom values for the Charente at this station during summer. In order to provide a better feel for the model performance, we added a figure of the seasonal variations for the stations closest to the marine boundary within the salinity gradient in the supplementary material (Figure S-3). It can be observed that, in the cross-section in Figure S-3, PO4 and DO values were better simulated than at the station presented in the main text although we acknowledge that NH4 was still too low.

In spite of the fact that the performances of the model are not as convincing in the Charente as they are in other systems, we believe it was nonetheless relevant to include them in our manuscript for two main reasons:

1-The performance of our model is the Charente, although not fully accurate, still falls within the range of values of the observations (see Figure 4) for most of the variables. Ammonium is indeed not well simulated by the model but its level is low compared to nitrate, which is well represented. We also acknowledge that the strong depletion in oxygen is not reproduced by the model. Overall it seems that the model can nonetheless provide information on the main biogeochemical characteristics of an estuary, even when data are scarce.

2-We would like to stress out the results of our simulations are generated by the application of the same biogeochemical parametrization to all systems. Rather than calibrating our model to each system individually, which would certainly yield better results but would essentially consist of developing 7 individual models. Instead, we wanted to evaluate how much insight can be provided by the application of a generic model and parametrization at the regional scale as a first step towards the application of such modeling approach to all the estuaries of a continuous stretch of coast, regardless of their data coverage. We thus believe that the arguably 'mediocre' performance of our model in the Charente and the ability of our simulations to capture the overall dynamics of 7 different systems with a single set-up is representative of the level of confidence we could have in results obtained for estuaries totally devoid of data. In this regard, we believe it is important to include the results of all our simulated systems and that this level of accuracy is very encouraging in the perspective of future regional simulations.

**RC3.11:** Many statements in the results section contain an opinion or assessment:

**AC3.11:** Thanks for the suggestions. Several statements were reconsidered.

Line 305: "These comparisons… available observations"

**AC2.11a:** The sentence has been rewritten and now reads:

*"The model outputs were compared to available observations in order to provide bias and RMSE, which are reported in Table 7. These statistical indicators reflect overall good performances of the model through the standards provided by Moriasi et al. (2007, −0.7 < Bias < 0.7)."*

Line 335: "seasonal trends are properly captured…"

**AC2.11b:** The sentence has been rewritten and now reads:

*"In addition, the seasonal trends generated by the model for nutrients, DO, and Chl-a concentrations generally follow the variations reported by field measurements both in timing and amplitude."*

Line 342-344: "DO showed… high summer mineralization": Please point to results that back up this statement about the relative importance of the effect of solubility and mineralization. Figure 4 only shows peaks and troughs.

Line 344: "Conversely, phytoplankton… time scale". Please, just describe the model output.

**AC2.11c:** A sentence has been added, and now the paragraph is as followed:

*"DO showed a regular trend with high values in winter and low values in summer, according to its solubility, but also to its consumption with high summer mineralization. Indeed, whereas the Charente River showed DO observed values of about 11 mg $L^{-1}$ in winter (~100% saturation) much lower DO values down to 3-4 mg $O_2$ $L^{-1}$ were found in summer (i.e. ~35% saturation, well illustrating a high summer $O_2$ consumption ( **Error! Reference source not found.**). Indeed, phytoplankton biomass (Chl a) simulations showed a shift in relation to the observations (with a right level, however), which led to short-term summer DO peaks of the model which did not fit the observations. Noteworthy is the excessively scarce phytoplankton data which cannot support the modelled pattern at such a time scale."*

Line 354-360: "Considering the suitable agreement… retention rates." The first sentence is an assessment and the remainder of the paragraph are methods.

**AC2.11d:** Many thanks for this remark, the following sentence has been removed from the results section and appears in method one**:** dissolved inorganic nitrogen (DIN=$NH_4$+$NO_3$), dissolved inorganic phosphorus (DIP=$PO_4$), dissolved silica (DSi), Phy (Chl-a, using a C/Chl-a ratio of 40 (Jakobsen and Markager, 2016)), and TOC were calculated (**Error! Reference source not found.**). In addition, the Redfield–Brzezinkski ratio C:N:P:Si = 106:16:1:15–20 (Brzezinski, 1985; Redfield et al., 1963) was used to take into account the organic fractions and estimate total nitrogen (TN), total phosphorus (TP), and total silica (TSi), whose fluxes were preferentially chosen for calculating overall retention rates (see section **Error! Reference source not found.**).

Line 412-414: "Most of the… retention itself"

**AC2.11f:** This sentence has also been removed from the results section: most of the retention may therefore occur in the dam reservoir, thus favoring nutrient uptake by phytoplankton and loss of N via denitrification (Garnier et al., 1999; Seitzinger et al., 2006; Yan et al., 2021) and in total reducing estuarine retention itself.

There are other examples where the tone is not neutral or the text is not limited to results. The authors could improve the text by revising the results section.

Several sentences have been shortened in the result section to ensure neutrality.

**Minor comments:**

**RC3.12:** Line 155: "Numerical Schemes". This paragraph mentions already existing models that have been used and describes the conceptual model, but not so much the numerical implementation. Therefore, consider changing the heading.

**AC3.12:** We followed the suggestion of the reviewer and updated the title of this section to "Model Description and Set-up".

**RC3.13:** Line 173 (Figure 2b): What process does the arrow from "Denitrification" to "Aerobic Degradation" represent?

**AC3.13:** This arrow did not correspond to any actual flux but was the result of a mistake in the design of the conceptual scheme. The figure was corrected and the new Figure 2b now accurately represents the different fluxes and state variables of the model.

[Figure]

Figure 1 (a) The C-GEM concept. (b) Conceptual scheme of the biogeochemical module used in C-GEM in this study: a circle represents the state variables while a rectangle represents the processes; Dia corresponds to diatoms.

**RC3.14:** Line 204: "spin-up" instead of "warm-up"

**AC3.14:** This correction was implemented.

**RC3.15:** Line 204-205: "repeating the… steady-state conditions." Here I got lost.

**AC3.15:** The sentence has been rewritten as follows:

*"The simulation starts following a 60-day spin-up during which only the hydrodynamics and transport modules are resolved over a repeating identical tidal cycle using, which enables the system to reach a dynamic steady-state, providing realistic initial conditions for the biogeochemical module."*

In a nutshell, the 60 days model spin-up is performed using only the hydrodynamics and transport module to generate realistic longitudinal profiles for all state variables before the beginning of the actual simulation. These concentration profiles provide the biogeochemical module with realistic initial conditions constrained by the physics of the system.

**RC3.16:** Line 209: "Calibration was implemented based on 2015, as average…" Consider rephrasing.

**AC3.16:** The sentence was changed into:

*"Calibration was implemented based on 2015, as it is representative of average hydrological conditions in France in recent years."*

**RC3.17:** Line 337-338: "showed a seasonal decrease from winter to autumn"… probably from autumn to winter is meant.

**AC3.17:** Yes indeed, we apologize for the mistake. The correction was made.

---

## Author Response (AR2)

Comments to the first revision of "Nutrient transport and transformation in macrotidal estuaries of the French Atlantic coast: a modelling approach using C-GEM" by Xi Wei et al.

In my previous review my main criticism was that the manuscript does not provide a description of the simulated processes that may give readers a mechanistic understanding of the biogeochemical dynamics in the studied estuaries. The authors responded by adding texts about which processes were or were not incorporated, adding the reaction network in the supplement, and elaborating on the numerics in the response to reviewers. However, this only describes the methods that are used. The results and discussion still do not elaborate on the rates of processes, which are resolved by the model.

We are thankful to the reviewer for acknowledging the additions and answers we provided in the previous round of reviews. Based on this second evaluation, we now understand that the reviewer was not actually asking for further explanations and description regarding the model implementation and formulation (which was actually what the other two reviewers were requesting) but rather a more in-depth analysis and discussion of the biogeochemical process simulated by our model and a quantification of those biogeochemical transformations. We certainly understand the interest of such discussion and provide in the updated manuscript a several additions to strengthen that aspect of our study. We would also like to point out that we did already provide quantifications of several biogeochemical processes in a new table during the previous revision but we acknowledge that we did not discuss these values in the manuscript.

In the following document, you will find point by point answers to comments of the reviewers and several additions to the manuscript and supplementary material that, hopefully, addresses the desire of the reviewer for more mechanistic analysis of our model results. The key additions to the manuscript include:

- A totally new section in the results entirely devoted to the quantification of the main biogeochemical transformation occurring in the studied estuaries.
- An updated of table S2 of the supplementary material following which is now better connected to the main manuscript as it supports the new section mentioned in the previous point
- Some simulations performed without the biogeochemical module have been carried out, following a reviewer's suggestion and a new figure was added to the supplementary material which compares mixing curves for TOC and NO3 between simulations ignoring or accounting for biogeochemical transformations.
- Time series for upstream and downstream boundary conditions for the main biogeochemical state variables as well as the fresh water discharge are now provided for all systems in order to provide a better feel for the magnitudes and characteristic time-scales of the variations of the conditions at both edges of the estuarine systems.
- The discussion of the limitations of the model and, in particular, the absence of explicit benthic module has been improved following the reviewer's remarks.

However, we felt like some of the reviewer's requests and suggestions, while certainly interesting, were getting past the scope of our study.

I tried to give suggestions to elaborate on the biogeochemistry. One suggestion was to conduct a sensitivity analysis. The authors refer for this to another paper (Volta et al., 2014) where the same model was applied, but as the forcing through the boundary conditions and the geometries are

different, the response to changing parameters in a non-linear model is likely also different. I also suggested a simulation with the biogeochemistry turned off, as this would allow the authors to visualize in Fig 4 to what extent temporal variations driven by changing boundary conditions are modulated by the internal biogeochemistry. The boundary conditions at the ocean and the river side are constantly changing. This forcing based on data extracted from ECO-MARS3D, REPHY, and NAIADES is not shown in the manuscript. The manuscript only presents a table with the annual mean values for boundary conditions (Table 4), but it does not show the temporal variability (e.g. tides and seasons). This additional simulation without biogeochemistry may also allow for a better assessment of the model validation when data is only available near the boundaries (see comments RC1.9, RC1.10, RC1.12). The authors may still want to consider other ways to visualize the biogeochemistry that takes place within the model; for instance, showing reaction rates, presenting mass balances with the effect various processes on nutrient concentrations, etc.

We acknowledge that the reviewer provided a number of suggestions to investigate the internal biogeochemical processing of carbon and nutrient within the simulated estuaries. One of the suggestions was to perform a sensitivity analysis. Here, the reviewer suggests that this sensitivity analysis could be carried out by varying the boundary conditions applied to the different systems as it would allow assessing the effects of changing upstream and downstream constraints on the internal biogeochemical processing within the estuary. While we cannot deny that such analysis would certainly be of interest, we feel like it could also constitute an entire study in itself. We agree however that the previous version of the manuscript only provided annually averaged values for the boundary conditions used to constrain our simulation and this was not enough for the dedicated reader to evaluate the effect of, for example, varying river discharges on the biogeochemical dynamics of the model.

We thus created a large new figure with time series for the concentrations of the main biogeochemical state variables applied at the upstream and downstream boundary conditions for all systems. This figure was added to the supplementary material and can also be found at the end of this answer. We believe this will be very useful for the reader to evaluate the effect of the seasonal variations in fresh water discharges and nutrients concentrations on carbon and nutrients dynamics summarized on figures 3 and 4 of the main manuscript and completed by figures S1-S3 of the supplementary material. Indeed, we provided several longitudinal profiles and time-series for all the main biogeochemical variables simulated by the model and we believe that the (now possible) comparison between the temporal evolution of time-series extracted within the model domain with the seasonal variations of the upstream boundary condition will provide an insight to the reader regarding the biogeochemical dynamics within the estuary.

We would also like to point out that our study purposely investigates very different systems characterized by very different upstream constrains, residence times and geometries. While this does not in itself qualify as a sensitivity analysis, this strategy was nonetheless designed to assess the effect of these differences in boundary conditions on the resulting fate of carbon and nitrogen within the simulated estuaries.

[Figure]

**Figure S-4 Temporal variations for salinity, fresh water discharge, nitrate (NO3), ammonium (NH4), dissolved silica (Si), phosphate (PO4), dissolved oxygen (DO), and total organic carbon (TOC) concentrations from 2014 to 2016 at the downstream (DBC) and upstream boundary conditions (UBC) for all the estuarine systems simulated in this study. Salinity is only provided at the DBC while fresh water discharge is only provided for the UBC.**

AC3.1 does not address my comment, which was about showing reaction rates and elaborating on the modeled rates. It was not about which processes are included in the model or the kinetic rate expressions.

We believe we now better understand what the reviewer was really asking for during the previous round of reviews. An entire new section was thus added to the results with a focus on the description and quantification of several biogeochemical processes (TOC degradation, denitrification, Si uptake) in order to better report the internal biogeochemical transformations

simulated by our model. The new section is also supported by a table summarizing these fluxes and reporting them to the riverine loads in order to better assess the respective magnitudes of the lateral transport and the biogeochemical transformations within the system.

*'The reaction process rates calculated by the model along the estuaries over the 3 years were integrated and provided as annual average, in ton of total organic Carbon, N-NO3, dissolved Si or phosphate per year for organic matter degradation, denitrification, net primary production either from diatoms or the from all the algal community, respectively (see Table S-1 in supplementary material). These fluxes are highest in the largest estuaries with the largest carbon and nutrients loads as well as the highest residence time. Reported in terms of percentage of the import fluxes, these values allow comparing the intensity of the biogeochemical processing simulated by the model between the different estuaries as well as with carbon and nutrients retention rates discussed above. Overall, Total organic carbon degradation and NO3 denitrification percentages were the highest in the Gironde (98 % and 26%, respectively) and the lowest for the Vilaine (10%, 3%, respectively). Other systems displayed intermediate values falling in the 24-39% and 3-14% ranges for TOC degradation and denitrification, respectively (Table S-1). This intensity of organic carbon processing is consistent with the global figures suggested at the global scale by Bauer et al. (2013) as well as previous modeling studies performed with C-GEM in Europe and along the East coast of the US (Volta et al., 2016, Laruelle et al., 2017, 2019). The integrated denitrification rates are also consistent with the compilation performed by Nixon et al (1996). Biogeochemical reactivity regarding organic matter degradation and denitrification appeared greater for the most retentive Gironde, with its longest residence time. This trend was also illustrated by the calculation of mixing curves for the Seine, Loire and the Gironde (the 3 systems with the longest salinity gradient) computed for the reference simulation and simulations in which the biogeochemical model of C-GEM was deactivated (Figure S2 of the supplementary material). The difference between the refence simulation and the simulation without biogeochemistry provides a visual representation of the intensity of the biogeochemical processing within the system and is significantly more pronounced in the Gironde. Interestingly, silica uptake percentages, which are entirely sustained by diatoms primary production were also the highest in the Gironde, the Seine and the Somme (16%, 13% and 12%, respectively), in accordance with the ones for phosphates (16%, 43%, 76%, respectively). These results revealed however a large range of autotrophic activity. The percentage PO4 uptake flux was particularly high in the Somme because of its proportionally low specific P riverine loads from its upstream boundary. Overall biogeochemical phosphates fluxes were rather well balanced with those of silica, according to the Redfield ratios, indicating mostly a development of diatoms (Table S-1).'*

AC3.2 does not address my comment. Here I asked for a sensitivity analysis of the model outcomes with regard to the biogeochemistry occurring within the model domain. It was not about the numerical implementation.

Again, we acknowledge that we did not initially understand what the reviewer was expecting but we believe the that new section of text presented above now addresses the desire of the reviewer for a better description of the inner biogeochemical dynamics of the model. We also note that the reviewer suggested to perform a fully blown sensitivity analysis of the model and we already justified why we feel like such an extensive analysis was not justified to and falls outside of the scope of this particular paper. We did however follow the suggestion of the reviewer to perform simulations in which the biogeochemical module of our model was deactivated. These simulations allowed us calculating mixing curves for TOC and NO3 for the Seine, Loire and the Gironde (the 3

systems with the longest salinity gradient) which were added to the supplementary material and referred to in the manuscript.

[Figure]

**Figure S-5 Mixing curves extracted from the model outputs for total organic carbon (TOC) and nitrate (NO3) along the Seine, Loire and Gironde estuaries on May 1st 2014 and November 1st 2014. Continuous lines correspond to simulations performed without the biogeochemical module of C-GEM while dashed lines correspond to the reference simulations.**

AC3.3 and RC/AC3.9:

The text at lines 514-524 does not provide a compelling argument not to include a benthic module.

Following the reviewer's comment, we modified this section of the discussion in the updated manuscript and tried to better express the fact that we do not argue that benthic processes should be ignored altogether but, rather, that the model set-up used for our study was first step (with a first order representation of burial) towards better representation of carbon and nutrient removal within estuaries, starting from generic model without any interaction with the sediments. The updated text now reads as follows:

'*With its current set-up, the lack of explicit benthic biogeochemical module obviously limits the depth of mechanistic understanding the model can provide of nutrient cycling, particularly regarding interactions between pelagic and benthic compartments which can significantly influence the intensity but also the timing of nutrients and organic matter cycling in estuaries (Laruelle et al., 2019). In that context, future developments of the model should include the implementation of several benthic processes this task comes with a number of hurdles. For instance, while the addition of a full diagenetic module at each grid cell of our model would be possible, it would also increase its calculation time by one order of magnitude and likely require a very long spin-up to generate initial conditions for the benthic species. There exist simpler benthic modules of lower complexity, which would limit the computation cost of adding an explicit representation of benthic processes (Billen et al., 2015; Soetaert et al., 2006) but those would nonetheless significantly increase the data demand of the model to be properly calibrated. Indeed, the increase in complexity of the model will involve the use of field data to constrain and calibrate the newly implemented processed. While measurements of estuarine benthic processing of nutrients and carbon do exist, they are still relatively scarce. In the present study, the simple representation of particulate matter burial that was implemented and applied to phytoplankton and TOC to provide a first-order representation of the process, which is necessary to evaluate the retention of carbon and nutrients within the system. We believe this addition, coupled with denitrification provides a first insight on the main pathways removing nutrients from estuaries and allows calculating carbon and nutrient retention rates that can be compared with previously published estimates.'*

If including benthic processes leads to more uncertainty (lines 519-520) then ignoring these processes cannot make the model outcomes more reliable. Letting water-column processes account for sedimentary processes makes no sense. For instance, as the water-column is fully oxygenated it will not allow for denitrification and cannot compensate for missing benthic denitrification.

We feel like the comment of the reviewer ('…benthic processes leads to more uncertainty…') does not accurately reflect what we were trying to express in lines 519-520 but acknowledge that our text might not have been clear enough. Our point was that an increase of the complexity of the model associated to the implementation of new fluxes requires new data for the calibration of the model. In the case of estuarine benthic processes, we do not deny that such data exist but they remain quite scarce. In any case, the sentence singled out by the reviewer was removed.

Moreover, while we understand and agree with the remark of the reviewer that 'Letting water-column processes account for sedimentary processes…' is not an ideal solution, we would also like to point out that the practice of simplifying the conceptual representation of nitrogen processing within estuaries (or rivers, lakes or reservoirs) is not new and can be found, for instance, in simple mechanistic models such as those of Maavara et al. (2019). Again, these approaches do not accurately represent benthic dynamics but are a compromise and an intermediate step between totally ignoring the benthos and using a model too complex to be adequately calibrated with available data.

Maavara, T, Lauerwald, R, Laruelle, GG, et al. Nitrous oxide emissions from inland waters: Are IPCC estimates too high? Glob Change Biol. 2019; 25: 473– 448. https://doi.org/10.1111/gcb.14504

If the authors do not want to include a benthic module for this paper, they should try to assess the effect of omitting benthic processes. Typical rates for benthic processes may be taken from literature and be compared to simulated and measured rates in the water column. The importance of benthic reactions can also be estimated by calculating Damköhler numbers.

We understand the point the reviewer is trying to make and we believe that our discussion of the current level of complexity of the reaction network of our biogeochemical model demonstrates our willingness to openly discuss the limitations of our approach. However, we are not convinced that assessing or quantifying the effect of omitting benthic processes on the results of our simulations can be easily performed using any of the method suggested by the reviewer. While there indeed exist observation and model-derived estimates for benthic process rates such as denitrification, remineralization or burial in some well-studied estuaries, those are typically very site-specific and widely vary from a system to the next. As a consequence, evaluating the potential magnitude of the 'omitted' processes on the basis of such uncertain data would likely yield fluxes with very larges uncertainties resulting in limited insight. Bearing in mind the limits of our approach, we feel like comparing the carbon and nutrient retention rates simulated by our model with previously published literature (as we did in our discussion) and evaluating our model outputs against available water-column measurements is arguably a more concrete (yet indirect) way of evaluating the potential effect of missing benthic processes on our simulations.

AC3.8: The abstract states that one of the goals of the paper is to constrain chemical budgets, not the comparison of models or model development. So is it not better to just increase the grid resolution?

In C-GEM, changing the spatial resolution of the grid is not as trivial a task as the reviewer seems to suggest and we still believe that the implementation of such modification would only have added little insight regarding the accuracy of our simulations considering the very limited amount of data available for the smallest estuaries (Somme and Vilaine in particular). Besides, in spite of what the reviewer seems to understand from reading our abstract, one of our goals was indeed apply the same model set-up to several estuarine systems along the French Atlantic coast and take full advantage of the genericity of C-GEM with as few as possible modifications from site to site (besides forcings and boundary conditions). To take into account the reviewer's remark and better convey this objective of our work, the following statement was added at the end of the abstract: *'This study also demonstrates the ability of our model to be applied with a similar set-up to several estuarine systems characterized by different size, geometries and riverine loads.'*